# Why Propagate Alone?
# Parallel Use of Labels and Features on Graphs

**Yangkun Wang**[1†]**, Jiarui Jin**[1†]**, Weinan Zhang**[1]**, Yongyi Yang**[2†]**, Jiuhai Chen**[3†]**, Quan Gan**[4]**,
Yong Yu**[1]**, Zheng Zhang**[4]**, Zengfeng Huang**[2]**, David Wipf**[4]
[1]Shanghai Jiao Tong University, [2]Fudan University, [3]University of Maryland, [4]Amazon
`daviwipf@amazon.com`

## Abstract

Graph neural networks (GNNs) and label propagation represent two interrelated modeling strategies designed to exploit graph structure in tasks such as node property prediction. The former is typically based on stacked message-passing layers that share neighborhood information to transform node features into predictive embeddings. In contrast, the latter involves spreading label information to unlabeled nodes via a parameter-free diffusion process, but operates independently of the node features. Given then that the material difference is merely whether features or labels are smoothed across the graph, it is natural to consider combinations of the two for improving performance. In this regard, we have recently proposed to use a randomly-selected portion of the training labels as GNN inputs, concatenated with the original node features for making predictions on the remaining labels. This so-called *label trick* accommodates the parallel use of features and labels, and is foundational to many of the top-ranking submissions on the Open Graph Benchmark (OGB) leaderboard. And yet despite its wide-spread adoption, thus far there has been little attempt to carefully unpack exactly what statistical properties the label trick introduces into the training pipeline, intended or otherwise. To this end, we prove that under certain simplifying assumptions, the stochastic label trick can be reduced to an interpretable, deterministic training objective composed of two factors. The first is a data-fitting term that naturally resolves potential label leakage issues, while the second serves as a regularization factor conditioned on graph structure that adapts to graph size and connectivity. Later, we leverage this perspective to motivate a broader range of label trick use cases, and provide experiments to verify the efficacy of these extensions.

## 1 Introduction

Node property prediction is a ubiquitous task involving graph data with node features and/or labels, with a wide range of instantiations across real-world scenarios such as node classification (Velickovic et al., 2018) and link prediction (Zhang & Chen, 2018), while also empowering graph classification (Gilmer et al., 2017), etc. Different from conventional machine learning problems where there is typically no explicit non-iid structure among samples, nodes are connected by pre-specified edges, and a natural assumption is that labels and features vary smoothly over the graph. This smoothing assumption has inspired two interrelated lines of research: First, graph neural networks (GNNs) (Kipf & Welling, 2017; Hamilton et al., 2017; Li et al., 2018; Xu et al., 2018; Liao et al., 2019; Xu et al., 2019) leverage a parameterized message passing strategy to convert the original node features into predictive embeddings that reflect the features of neighboring nodes. However, this approach does not directly utilize existing label information beyond their influence on model parameters through training. Second, label propagation algorithms (LPA) (Zhu, 2005; Zhou et al., 2003; Zhang & Lee, 2006; Wang & Zhang, 2006; Karasuyama & Mamitsuka, 2013; Gong et al., 2017; Liu et al., 2019) spread label information via graph diffusion to make predictions, but cannot exploit node features.

As GNNs follow a similar propagation mechanism as the label propagation algorithm, the principal difference being whether features or labels are smoothed across the graph, it is natural to consider combinations of the two for improving performance. Examples motivated by this intuition, at least to varying degrees, include APPNP (Klicpera et al., 2019), Correct and Smooth (C&S) (Huang et al.,

---

[†]Work done during internship at Amazon Web Services Shanghai AI Lab.

2021), GCN-LPA (Wang & Leskovec, 2020), and TPN (Liu et al., 2019). While often effective, these methods are not all end-to-end trainable and easily paired with arbitrary GNN architectures. And in a related fashion Jia & Benson (2021) introduce an elegant generative framework that unifies LPA and GNNs, although the parallel propagation of both features and labels is not explicitly considered.

Among these various combination strategies, our previously proposed *label trick* (Wang et al., 2021) has enjoyed widespread success in facilitating the parallel use of node features and labels via a stochastic label splitting technique. In brief, the basic idea is to use a randomly-selected portion of the training labels as GNN inputs, concatenated with the original node features for making predictions on the remaining labels during each mini-batch. The ubiquity of this simple label trick is evidenced by its adoption across numerous GNN architectures and graph benchmarks (Sun & Wu, 2020; Kong et al., 2020; Shi et al., 2021; Li et al., 2021). And with respect to practical performance, this technique is foundational to many of the top-ranking submissions on the Open Graph Benchmark (OGB) leaderboard (Hu et al., 2020). For example, at the time of this submission, the top 10 results spanning multiple research teams all rely on the label trick, as do the top 3 results from the recent KDDCUP 2021 Large-Scale Challenge MAG240M-LSC (Hu et al., 2021).

And yet despite its far-reaching adoption, thus far the label trick has been motivated primarily as a training heuristic without a strong theoretical foundation. Moreover, many aspects of its underlying operational behavior have not been explored, with non-trivial open questions remaining. For example, while originally motivated from a stochastic perspective, is the label trick reducible to a more transparent deterministic form that is amenable to interpretation and analysis? Similarly, are there any indirect regularization effects with desirable (or possibly undesirable) downstream consequences? And how do the implicit predictions applied by the model to test nodes during the stochastic training process compare with the actual deterministic predictions used during inference? If there is a discrepancy, then the generalization ability of the model could be compromised. And finally, are there natural use cases for the label trick that have so far flown under the radar and been missed? We take a step towards answering these questions via the following two primary contributions:

- We prove that in certain restricted settings, the original *stochastic* label trick can be reduced to an interpretable, *deterministic* training objective composed of two terms: (1) a data-fitting term that naturally resolves potential label leakage issues and maintains consistent predictions during training and inference, and (2) a regularization factor conditioned on graph structure that adapts to graph size and connectivity. Furthermore, complementary experiments applying the label trick to a broader class of graph neural network models corroborate that similar effects exists in more practical real-world settings, consistent with our theoretical findings.

- Although in prior work the label trick has already been integrated within a wide variety of GNN models, we introduce novel use-cases motivated by our analysis. This includes exploiting the label trick to: (i) train simple end-to-end variants of label propagation and C&S, and (ii) replace stochastic use cases of the label trick with more stable, deterministic analogues that are applicable to GNN models with linear propagation layers such as SGC (Wu et al., 2019), TWIRLS (Yang et al., 2021) and SIGN (Frasca et al., 2020). Empirical results on node classification benchmarks verify the efficacy of these simple enhancements.

Collectively, these efforts establish a more sturdy foundation for the label trick, and in doing so, help to ensure that it is not underutilized.

## 2 BACKGROUND

Consider an undirected graph $G = (V, E)$ with $n = |V|$ nodes, the node feature matrix is denoted by $\boldsymbol{X} \in \mathbb{R}^{n \times d}$ and the label matrix of the nodes is denoted by $\boldsymbol{Y} \in \mathbb{R}^{n \times c}$, with $d$ and $c$ being the number of channels of features and labels, respectively. Let $\boldsymbol{A}$ be the (unweighted) adjacency matrix, $\boldsymbol{D}$ the degree matrix and $\boldsymbol{S} = \boldsymbol{D}^{-\frac{1}{2}} \boldsymbol{A} \boldsymbol{D}^{-\frac{1}{2}}$ the symmetric normalized adjacency matrix. The symmetric normalized Laplacian $\boldsymbol{L}$ can then be formulated as $\boldsymbol{L} = \boldsymbol{I}_n - \boldsymbol{S}$. We also define a training mask matrix as $\boldsymbol{M}_{tr} = \begin{pmatrix} \boldsymbol{I}_m & \boldsymbol{0} \\ \boldsymbol{0} & \boldsymbol{0} \end{pmatrix}_{n \times n}$, where w.l.o.g. we are assuming that the first $m$ nodes, denoted $\mathcal{D}_{tr}$, form the training dataset. We use $\boldsymbol{P}$ to denote a *propagation matrix*, where the specific $\boldsymbol{P}$ will be described in each context.

## 2.1 LABEL PROPAGATION ALGORITHM

Label propagation is a semi-supervised algorithm that predicts unlabeled nodes by propagating the observed labels across the edges of the graph, with the underlying smoothness assumption that two nodes connected by an edge are likely to share the same label. Following Zhou et al. (2003); Yang et al. (2021), the implicit energy function of label propagation is given by

$$E(\boldsymbol{F}) = (1 - \lambda)\|\boldsymbol{F} - \boldsymbol{Y}_{tr}\|_2^2 + \lambda \operatorname{tr}[\boldsymbol{F}^\top \boldsymbol{L} \boldsymbol{F}], \tag{1}$$

where $\boldsymbol{F}$ is the smoothed labels, $\boldsymbol{Y}_{tr} = \boldsymbol{M}_{tr}\boldsymbol{Y}$ is the label matrix of training nodes, and $\lambda \in (0, 1)$ is a regularization coefficient that determines the trade-off between the two terms. The first term is a *fitting constraint*, with the intuition that the predictions of a good classifier should remain close to the initial label assignments, while the second term introduces a *smoothness constraint*, which favors similar predictions between neighboring nodes in the graph.

It is not hard to derive that the closed-formed optimal solution of this energy function is given by $\boldsymbol{F}^* = \boldsymbol{P}\boldsymbol{Y}$, where $\boldsymbol{P} = (1 - \lambda)(\boldsymbol{I}_n - \lambda \boldsymbol{S})^{-1}$. However, since the stated inverse is impractical to compute for large graphs, $\boldsymbol{P}\boldsymbol{Y}$ is often approximated in practice via $\boldsymbol{P} \approx (1 - \lambda) \sum_{i=0}^k \lambda^i \boldsymbol{S}^i \boldsymbol{Y}$. From this expression, it follows that $\boldsymbol{F}$ can be estimated by the more efficient iterations $\boldsymbol{F}^{(k+1)} = \lambda \boldsymbol{S}\boldsymbol{F}^{(k)} + (1 - \lambda)\boldsymbol{F}^{(0)}$, where $\boldsymbol{F}^{(0)} = \boldsymbol{Y}_{tr}$ and for each $k$, $\boldsymbol{S}$ smooths the training labels across the edges of the graph.

## 2.2 GRAPH NEURAL NETWORKS FOR PROPAGATING NODE FEATURES

In contrast to the propagation of labels across the graph, GNN models transform and propagate node features using a series of feed-forward neural network layers. Popular examples include GCN (Kipf & Welling, 2017), GraphSAGE (Hamilton et al., 2017), GAT (Velickovic et al., 2018), and GIN (Xu et al., 2019). For instance, the layer-wise propagation rule of GCN can be formulated as $\boldsymbol{X}^{(k+1)} = \sigma(\boldsymbol{S}\boldsymbol{X}^{(k)}\boldsymbol{W}^{(k)})$ where $\sigma(\cdot)$ is an activation function such as ReLU, $\boldsymbol{X}^{(k)}$ is the $k$-th layer node representations with $\boldsymbol{X}^{(0)} = \boldsymbol{X}$, and $\boldsymbol{W}^{(k)}$ is a trainable weight matrix of the $k$-th layer. Compared with label propagation, GNNs can sometimes exhibit a more powerful generalization capability via the interaction between discriminative node features and trainable weights.

## 2.3 COMBINING LABEL AND FEATURE PROPAGATION

While performing satisfactorily in many circumstances, GNNs only indirectly incorporate ground-truth training labels via their influence on the learned model weights. But these labels are not actually used during inference, which can potentially degrade performance relative to label propagation, especially when the node features are noisy or unreliable. Therefore, it is natural to consider the combination of label *and* feature propagation to synergistically exploit the benefits of both as has been proposed in Klicpera et al. (2019); Liu et al. (2019); Wang & Leskovec (2020); Wang et al. (2021); Shi et al. (2021); Huang et al. (2021).

One of the most successful among these hybrid methods is our previously proposed *label trick* (Wang et al., 2021), which can be conveniently retrofitted within most standard GNN architectures while facilitating the parallel propagation of labels and features in an end-to-end trainable fashion. As mentioned previously, a number of top-performing GNN pipelines have already adopted this trick, which serves to establish its widespread relevance (Sun & Wu, 2020; Kong et al., 2020; Li et al., 2021; Shi et al., 2021) and motivates our investigation of its properties herein. To this end, we formally define the label trick as follows:

**Definition 1 (label trick)** *The label trick is based on creating random partitions of the training data as in $\mathcal{D}_{tr} = \mathcal{D}_{in} \cup \mathcal{D}_{out}$ and $\mathcal{D}_{in} \cap \mathcal{D}_{out} = \varnothing$, where node labels from $\mathcal{D}_{in}$ are concatenated with the original features $\boldsymbol{X}$ and provided as GNN inputs (for nodes not in $\mathcal{D}_{in}$ zero-padding is used), while the labels from $\mathcal{D}_{out}$ serve in the traditional role as supervision. The resulting training objective then becomes*

$$\mathop{\mathbb{E}}_{splits} \left[ \sum_{i \in \mathcal{D}_{out}} \ell\big( \boldsymbol{y}_i, \ f[\boldsymbol{X}, \boldsymbol{Y}_{in}; \mathcal{W}]_i \big) \right] \tag{2}$$

*where $\boldsymbol{Y}_{in} \in \mathbb{R}^{n \times c}$, is defined row-wise as $\boldsymbol{y}_{in,i} = \begin{cases} \boldsymbol{y}_i & \text{if } i \in \mathcal{D}_{in} \\ 0 & \text{otherwise} \end{cases}$ for all $i$, the function $f(\boldsymbol{X}, \boldsymbol{Y}_{in}; \mathcal{W})$ represents a message-passing neural network with parameters $\mathcal{W}$ and the concatenation of $\boldsymbol{X}$ and $\boldsymbol{Y}_{in}$ as inputs, and $\ell(\cdot, \cdot)$ denotes a point-wise loss function over one sample/node. At inference time, we then use the deterministic predictor $f[\boldsymbol{X}, \boldsymbol{Y}_{tr}; \mathcal{W}]_i$ for all test nodes $i \notin \mathcal{D}_{tr}$.*

## 3   RELIABLE RANDOMNESS THOUGH THE LABEL TRICK

Despite its widespread adoption, the label trick has thus far been motivated as merely a training heuristic without formal justification. To address this issue, we will now attempt to quantify the induced regularization effect that naturally emerges when using the label trick. However, since the formal analysis of deep networks is challenging, we initially adopt the simplifying assumption that the function $f$ from (2) is linear, analogous to the popular SGC model from Wu et al. (2019). For simplicity of exposition, in Sections 3.1 and 3.2 we will consider the case where no node features are present to isolate label-trick-specific phenomena. Later in Sections 3.3 and 3.4 we will reintroduce node features to present our general results, as well as considering nonlinear extensions.

### 3.1   LABEL TRICK WITHOUT NODE FEATURES

Assuming no node features $\boldsymbol{X}$, we begin by considering the deterministic node label loss

$$\mathcal{L}(\boldsymbol{W}) = \sum_{i \in \mathcal{D}_{tr}} \ell\big(\boldsymbol{y}_i, \, [\boldsymbol{PY}_{tr}\boldsymbol{W}]_i\big), \tag{3}$$

where $[\,\cdot\,]_i$ indicates the $i$-th row of a matrix $\boldsymbol{PY}_{tr}\boldsymbol{W}$ is a linear predictor akin to SGC, but with only the zero-padded training label matrix $\boldsymbol{Y}_{tr}$ as an input. Additionally, $\boldsymbol{P}$ here can in principle be any reasonable propagation matrix, not necessarily the one associated with the original label propagation algorithm. However, directly employing (3) for training suffers from potential label leakage issues given that a simple identity mapping suffices to achieve the minimal loss at the expense of accurate generalization to test nodes. Furthermore, there exists an inherent asymmetry between the predictions computed for training nodes, where the corresponding labels are also used as inputs to the model, and the predictions for testing nodes where no labels are available.

At a conceptual level, these issues can be resolved by the label trick, in which case we introduce random splits of $\mathcal{D}_{tr}$ and modify (3) to

$$\mathcal{L}(\boldsymbol{W}) = \mathop{\mathbb{E}}_{splits} \Big[ \sum_{i \in \mathcal{D}_{out}} \ell\big(\boldsymbol{y}_i, \, [\boldsymbol{PY}_{in}\boldsymbol{W}]_i\big) \Big]. \tag{4}$$

For each random split, the resulting predictor only includes the label information of $\mathcal{D}_{in}$ (through $\boldsymbol{Y}_{in}$), and thus there is no unresolved label leakage issue when predicting the labels of $\mathcal{D}_{out}$. In practice, we typically sample the splits by assigning a given node to $\mathcal{D}_{in}$ with some probability $\alpha \in (0,1)$; otherwise the node is set to $\mathcal{D}_{out}$. It then follows that $\mathbb{E}[|\mathcal{D}_{in}|] = \alpha|\mathcal{D}_{tr}|$ and $\mathbb{E}[\boldsymbol{Y}_{in}] = \alpha\boldsymbol{Y}_{tr}$.

### 3.2   SELF-EXCLUDED SIMPLIFICATION OF THE LABEL TRICK

Because there exists an exponential number of different possible random splits, for analysis purposes (with later practical benefits as well) we first consider a simplified one-versus-all case whereby we enforce that $|\mathcal{D}_{out}| = 1$ across all random splits, with each node landing with equal probability in $\mathcal{D}_{out}$. In this situation, the objective function from (4) can be re-expressed more transparently without any expectation as

$$\mathcal{L}(\boldsymbol{W}) = \mathop{\mathbb{E}}_{splits} \Big[ \sum_{i \in \mathcal{D}_{out}} \ell\big(\boldsymbol{y}_i, \, [\boldsymbol{PY}_{in}\boldsymbol{W}]_i\big) \Big] = \mathop{\mathbb{E}}_{splits} \Big[ \sum_{i \in \mathcal{D}_{out}} \ell\big(\boldsymbol{y}_i, \, [\boldsymbol{P}(\boldsymbol{Y}_{tr} - \boldsymbol{Y}_i)\boldsymbol{W}]_i\big) \Big]$$
$$= \frac{1}{|\mathcal{D}_{tr}|} \sum_{i \in \mathcal{D}_{tr}} \ell\big(\boldsymbol{y}_i, \, [(\boldsymbol{P} - \boldsymbol{C})\boldsymbol{Y}_{tr}\boldsymbol{W}]_i\big), \tag{5}$$

where $\boldsymbol{Y}_i$ represents a matrix that shares the $i$-th row of $\boldsymbol{Y}$ and pads the rest with zeros, and $\boldsymbol{C} = \mathrm{diag}(\boldsymbol{P})$. This then motivates the revised predictor given by

$$f(\boldsymbol{Y}_{tr}; \boldsymbol{W}) = (\boldsymbol{P} - \boldsymbol{C})\boldsymbol{Y}_{tr}\boldsymbol{W}. \tag{6}$$

**Remark 1** *From this expression we observe the intuitive role that $\boldsymbol{C}$ plays in blocking the direct pathway between each training node input label to the output predicted label for that same node. In this way the predictor propagates the labels of each training node excluding itself, and for both training and testing nodes alike, the predicted label of a node is only a function of other node labels. This resolves the asymmetry mentioned previously with respect to the predictions from (3).*

**Remark 2** *It is generally desirable that a candidate model produces the same predictions on test nodes during training and inference to better ensure proper generalization. Fortunately, this is in fact the case when applying (6), which on test nodes makes the same predictions as label propagation. To see this, note that $\boldsymbol{M}_{te}(\boldsymbol{P} - \boldsymbol{C})\boldsymbol{Y}_{tr} = \boldsymbol{M}_{te}\boldsymbol{PY}_{tr}$, where $\boldsymbol{M}_{te} = \boldsymbol{I}_n - \boldsymbol{M}_{tr}$ is the diagonal mask matrix of test nodes and $\boldsymbol{PY}_{tr}$ is the original label propagation predictor.*

Although ultimately (6) will serve as a useful analysis tool below, it is also possible to adopt this predictor in certain practical settings. In this regard, $C$ can be easily computed with the same computational complexity as is needed to approximate $P$ as discussed in Section 2.1 (and for alternative propagation operators that are available explicitly, e.g., the normalized adjacency matrix, $C$ is directly available).

### 3.3 FULL EXECUTION OF THE LABEL TRICK AS A REGULARIZER

We are now positioned to extend the self-excluded simplification of the label trick to full execution with arbitrary random sampling, as well as later, the reintroduction of node features. For this purpose, we first define $\boldsymbol{Y}_{out} = \boldsymbol{Y}_{tr} - \boldsymbol{Y}_{in}$, and also rescale by a factor of $1/\alpha$ to produce $\widetilde{\boldsymbol{Y}}_{in} = \boldsymbol{Y}_{in}/\alpha$. The latter allows us to maintain a consistent mean and variance of the predictor across different sampling probabilities.

Assuming a mean square error (MSE) loss as computed for each node via $\ell(\boldsymbol{y}, \widehat{\boldsymbol{y}}) = ||\boldsymbol{y} - \widehat{\boldsymbol{y}}||_2^2$ (later we consider categorical cross-entropy), our overall objective is to minimize

$$\mathcal{L}(\boldsymbol{W}) = \mathop{\mathbb{E}}_{splits} \Big[ \sum_{i \in \mathcal{D}_{out}} \ell(\boldsymbol{y}_i, [\boldsymbol{P}\widetilde{\boldsymbol{Y}}_{in}\boldsymbol{W}]_i) \Big] = \mathop{\mathbb{E}}_{splits} \Big[ ||\boldsymbol{Y}_{out} - \boldsymbol{M}_{out}\boldsymbol{P}\widetilde{\boldsymbol{Y}}_{in}\boldsymbol{W}||_F^2 \Big], \quad (7)$$

where $\boldsymbol{M}_{out}$ is a diagonal mask matrix defined such that $\boldsymbol{Y}_{out} = \boldsymbol{M}_{out}\boldsymbol{Y} = \boldsymbol{Y}_{tr} - \boldsymbol{Y}_{in}$ and the random splits follow a node-wise Bernoulli distribution with parameter $\alpha$ as discussed previously. We then have the following:

**Theorem 1** *Define* $\boldsymbol{\Gamma} = \big( \mathrm{diag}(\boldsymbol{P}^T\boldsymbol{P}) - \boldsymbol{C}^T\boldsymbol{C} \big)^{\frac{1}{2}} \boldsymbol{Y}_{tr}$. *Then the label trick loss from (7) satisfies*

$$\frac{1}{1-\alpha} \mathop{\mathbb{E}}_{splits} \Big[ ||\boldsymbol{Y}_{out} - \boldsymbol{M}_{out}\boldsymbol{P}\widetilde{\boldsymbol{Y}}_{in}\boldsymbol{W}||_F^2 \Big] = ||\boldsymbol{Y}_{tr} - \boldsymbol{M}_{tr}(\boldsymbol{P} - \boldsymbol{C})\boldsymbol{Y}_{tr}\boldsymbol{W}||_F^2 + \frac{1-\alpha}{\alpha} ||\boldsymbol{\Gamma}\boldsymbol{W}||_F^2. \quad (8)$$

Note that $(\mathrm{diag}(\boldsymbol{P}^T\boldsymbol{P}) - \boldsymbol{C}^T\boldsymbol{C})$ is a positive semi-definite diagonal matrix, and hence its real square root will always exist. Furthermore, we can extend this analysis to include node features by incorporating the SGC-like linear predictor $\boldsymbol{P}\boldsymbol{X}\boldsymbol{W}_x$ such that Theorem 1 can then naturally be generalized as follows:

**Corollary 1** *Under the same conditions as Theorem 1, if we add the node feature term* $\boldsymbol{P}\boldsymbol{X}\boldsymbol{W}_x$ *to the label-based predictor from (7), we have that*

$$\frac{1}{1-\alpha} \mathop{\mathbb{E}}_{splits} \Big[ ||\boldsymbol{Y}_{out} - \boldsymbol{M}_{out}\boldsymbol{P}\boldsymbol{X}\boldsymbol{W}_x - \boldsymbol{M}_{out}\boldsymbol{P}\widetilde{\boldsymbol{Y}}_{in}\boldsymbol{W}_y||_F^2 \Big]$$
$$= ||\boldsymbol{Y}_{tr} - \boldsymbol{M}_{tr}\boldsymbol{P}\boldsymbol{X}\boldsymbol{W}_x - \boldsymbol{M}_{tr}(\boldsymbol{P} - \boldsymbol{C})\boldsymbol{Y}_{tr}\boldsymbol{W}_y||_F^2 + \frac{1-\alpha}{\alpha} ||\boldsymbol{\Gamma}\boldsymbol{W}_y||_F^2. \quad (9)$$

The details of the proofs of Theorem 1 and Corollary 1 are provided in Appendix B.1. This then effectively leads to the more general, feature and label aware predictor

$$f(\boldsymbol{X}, \boldsymbol{Y}_{tr}; \mathcal{W}) = \boldsymbol{P}\boldsymbol{X}\boldsymbol{W}_x + (\boldsymbol{P} - \boldsymbol{C})\boldsymbol{Y}_{tr}\boldsymbol{W}_y, \quad (10)$$

where $\mathcal{W} = \{\boldsymbol{W}_x, \boldsymbol{W}_y\}$. These theoretical results reveal a number of interesting properties regarding how the label trick behaves, which we summarize as follows:

**Remark 3** *Although the original loss involves an expectation over random data splits that is somewhat difficult to interpret, based on (9), the label trick can be interpreted as inducing a deterministic objective composed of two terms: (i) The error accrued when combining the original node features with the self-excluded label propagation predictor from (6) to mitigate label leakage, and (ii) An additional graph-dependent regularization factor on the model weights associated with the labels that depends on $\alpha$ (more on this below). We can also easily verify from (10) that the model applies the same prediction to test nodes during both training and inference, consistent with Remark 2.*

**Remark 4** *If the graph has no edges, then there is no chance for overfitting to labels and $\big( \mathrm{diag}(\boldsymbol{P}^T\boldsymbol{P}) - \boldsymbol{C}^T\boldsymbol{C} \big)^{\frac{1}{2}} = \boldsymbol{0}$ shuts off the superfluous regularization. In contrast, for a fully connected graph, the value of $\boldsymbol{\Gamma}$ can be significantly larger, which can potentially provide a beneficial regularization effect. Additionally, given that $\boldsymbol{\Gamma}$ also grows larger with graph size (assuming edges grow as well), $||\boldsymbol{\Gamma}\boldsymbol{W}_y||_F^2$ scales proportionately with the data fitting term, which is generally expected to increase linearly with the number of nodes. Hence (9) is naturally balanced to problem size.*

**Remark 5** *The splitting probability $\alpha$ in (9) controls the regularization strength. Specifically, when $\alpha$ tends to zero, fewer labels are used as input to predict a large number of output labels, which may be less reliable, and corresponds with adding a larger regularization effect. Additionally, it means placing more emphasis on the original node features and downplaying the importance of the labels as input in (9), which explains the addition of a penalty on $\boldsymbol{W}_y$. Conversely, when $\alpha$ tends to one, more labels are used as input to predict the output and the model approaches the deterministic self-excluded label trick. Specifically, for random splits where $|\mathcal{D}_{out}| = 1$, the loss mimics one random term from the self-excluded label trick summation, while for the splits when $|\mathcal{D}_{out}| = 0$, the contribution to the expectation is zero and therefore does not influence the loss. Splits with $|\mathcal{D}_{out}| > 1$ will have very low probability, so this situation naturally corresponds with canceling out the regularization term. Later in Section 3.4 we will extend these observations to general nonlinear models.*

We now turn to the categorical cross-entropy loss, which is more commonly applied to node classification problems. While we can no longer compute closed-form simplifications as we could with MSE, it is nonetheless possible to show that the resulting objective when using the original label trick is an upper bound on the analogous objective from the self-excluded label trick. More specifically, we have the following (see Appendix B.3 for the proof):

**Theorem 2** *Under the same conditions as in Theorem 1 and Corollary 1, if we replace the MSE loss with categorical cross-entropy we obtain the bound*

$$\frac{1}{1-\alpha} \mathop{\mathbb{E}}_{splits} \left[ \mathrm{CE}_{\mathcal{D}_{out}}(\boldsymbol{Y}_{out}, \boldsymbol{PXW}_x + \boldsymbol{P}\tilde{\boldsymbol{Y}}_{in}\boldsymbol{W}_y) \right] \geq \mathrm{CE}_{\mathcal{D}_{tr}}(\boldsymbol{Y}_{tr}, \boldsymbol{PXW}_x + (\boldsymbol{P}-\boldsymbol{C})\boldsymbol{Y}_{tr}\boldsymbol{W}_y),$$

(11)

*where $\mathrm{CE}_S(\cdot, \cdot)$ denotes the sum of row-wise cross-entropy of $S$.*

### 3.4 Nonlinear Extensions

When we move towards more complex GNN models with arbitrary nonlinear interactions, it is no longer feasible to establish explicit, deterministic functional equivalents of the label trick for general $\alpha$. However, we can still at least elucidate the situation at the two extremes where $\alpha \to 0$ or $\alpha \to 1$ alluded to in Remark 5. Regarding the former, clearly with probability approaching one, $\boldsymbol{Y}_{in}$ will always equal zero and hence the model will default to a regular GNN, effectively involving no label information as an input. In contrast, for the latter we provide the following:

**Theorem 3** *Let $f_{GNN}(\boldsymbol{X}, \boldsymbol{Y}; \boldsymbol{\mathcal{W}})$ denote an arbitrary GNN model with concatenated inputs $\boldsymbol{X}$ and $\boldsymbol{Y}$, and $\ell(\boldsymbol{y}, \hat{\boldsymbol{y}})$ a training loss such that $\sum_{i \in \mathcal{D}_{out}} \ell(\boldsymbol{y}_i, f_{GNN}[\boldsymbol{X}, \boldsymbol{Y}_{in}; \boldsymbol{\mathcal{W}}]_i)$ is bounded for all $\mathcal{D}_{out}$. It then follows that*

$$\lim_{\alpha \to 1} \left\{ \frac{1}{1-\alpha} \mathop{\mathbb{E}}_{splits} \left[ \sum_{i \in \mathcal{D}_{out}} \ell(\boldsymbol{y}_i, f_{GNN}[\boldsymbol{X}, \boldsymbol{Y}_{in}; \boldsymbol{\mathcal{W}}]_i) \right] \right\} = \sum_{i=1}^{m} \ell(\boldsymbol{y}_i, f_{GNN}[\boldsymbol{X}, \boldsymbol{Y}_{tr}-\boldsymbol{Y}_i; \boldsymbol{\mathcal{W}}]_i).$$

(12)

The proof is given in Appendix B.4. This result can be viewed as a natural generalization of (5), with one minor caveat: we can no longer guarantee that the predictor implicitly applied to test nodes during training will exactly match the explicit function $f_{GNN}[\boldsymbol{X}, \boldsymbol{Y}_{tr}; \boldsymbol{\mathcal{W}}]$ applied at inference time. Indeed, each $f_{GNN}[\boldsymbol{X}, \boldsymbol{Y}_{tr} - \boldsymbol{Y}_i; \boldsymbol{\mathcal{W}}]_i$ will generally produce slightly different predictions for all test nodes depending on $i$ unless $f_{GNN}$ is linear. But in practice this is unlikely to be consequential.

## 4 Broader Use Cases of the Label Trick

Although the label trick has already been integrated within a wide variety of GNN pipelines, in this section we introduce three novel use-cases motivated by our analysis.

### 4.1 Trainable Label Propagation

In Sections 3.1 and 3.2 we excluded the use of node features to simplify the exposition of the label trick; however, analytical points aside, the presented methodology can also be useful in and of itself for facilitating a simple, trainable label propagation baseline when we choose $\boldsymbol{P}$ as in Section 2.1.

The original label propagation algorithm from Zhou et al. (2003) is motivated as a parameter-free, deterministic mapping from a training label set to predictions across the entire graph. However, clearly the randomized label trick from Section 3.1, or its deterministic simplification from Section

3.2 can be adopted to learn a label propagation weight matrix $\boldsymbol{W}$. The latter represents a reasonable enhancement that can potentially help to compensate for interrelated class labels that may arise, especially in multi-label settings. In contrast, the original label propagation algorithm implicitly assumes that different classes are independent. Beyond this, other entry points for adding trainable weights are also feasible such as node-dependent or edge-dependent weights (Wang & Leskovec, 2020), nonlinear weighted mappings (Kipf & Welling, 2017), step-wise weights for heterophily graphs (Yamaguchi et al., 2016), or weights for different node types for heterogeneous graphs (Schlichtkrull et al., 2018).

## 4.2 DETERMINISTIC APPLICATION TO GNNs WITH LINEAR PROPAGATION LAYERS

Many prevailing GNN models follow the architecture of message passing neural networks (MPNNs). Among these are efficient variants that share node embeddings only through linear propagation layers. Representative examples include SGC (Wu et al., 2019), SIGN (Frasca et al., 2020) and TWIRLS (Yang et al., 2021). We now show how to apply the deterministic label trick algorithm as introduced in Section 3.2 with the aforementioned GNN methods.

We begin with a linear SGC model. In this case, we can compute $(\boldsymbol{P} - \boldsymbol{C})\boldsymbol{Y}_{tr}$ beforehand as the self-excluded label information and then train the resulting features individually without graph information, while avoiding label leakage problems. And if desired, we can also concatenate with the original node features. In this way, we have an algorithm that minimizes an energy function involving both labels and input features.

Additionally, for more complex situations where the propagation layers are not at the beginning of the model, the predictor can be more complicated such as $f(\boldsymbol{X}, \boldsymbol{Y}; \boldsymbol{W}) =$

$$h_1\Big(\sum_i \big[\boldsymbol{\mathcal{P}} h_0([\boldsymbol{X}, \boldsymbol{Y} - \boldsymbol{Y}_i])\big]_i\Big) = h_1(\boldsymbol{\mathcal{P}} h_0([\boldsymbol{X}, \boldsymbol{Y}]) - \boldsymbol{\mathcal{C}} h_0([\boldsymbol{X}, \boldsymbol{Y}]) + \boldsymbol{\mathcal{C}} h_0([\boldsymbol{X}, \boldsymbol{0}])), \quad (13)$$

where $[\cdot, \overset{i}{\cdot}]$ denotes the concatenation operation, $\boldsymbol{\mathcal{P}} = [\boldsymbol{P}_0, \boldsymbol{P}_1, \ldots, \boldsymbol{P}_{k-1}]^T$ is the integrated propagation matrix, $\boldsymbol{\mathcal{C}} = [\text{diag}(\boldsymbol{P}_0), \text{diag}(\boldsymbol{P}_1), \ldots, \text{diag}(\boldsymbol{P}_{k-1})]^T$, $h_0$ and $h_1$ can be arbitrary node-independent functions, typically multi-layer perceptrons (MLPs).

## 4.3 TRAINABLE CORRECT AND SMOOTH

Correct and Smooth (C&S) (Huang et al., 2021) is a simple yet powerful method which consists of multiple stages. A prediction matrix $\tilde{\boldsymbol{Y}}$ is first obtained whose rows correspond with a prediction from a shallow node-wise model. $\tilde{\boldsymbol{Y}}$ is subsequently modified via two post-processing steps, *correct* and *smooth*, using two propagation matrices $\{\boldsymbol{P}_c, \boldsymbol{P}_s\}$, where typically $\boldsymbol{P}_i = (1 - \lambda_i)(\boldsymbol{I}_n - \lambda_i \boldsymbol{S})^{-1}, i \in \{c, s\}$. For the former, we compute the difference between the ground truth and predictions on the training set as $\boldsymbol{E} = \boldsymbol{Y}_{tr} - \tilde{\boldsymbol{Y}}_{tr}$ and then form $\tilde{\boldsymbol{E}} = \gamma(\boldsymbol{P}_c \boldsymbol{E})$ as the *correction matrix*, where $\gamma(\cdot)$ is some row-independent scaling function. The final *smoothed* prediction is formed as $f_{C\&S}(\tilde{\boldsymbol{Y}}) = \boldsymbol{P}_s(\boldsymbol{Y}_{tr} + \boldsymbol{M}_{te}(\tilde{\boldsymbol{Y}} + \tilde{\boldsymbol{E}}))$. This formulation is not directly amenable to end-to-end training because of label leakage issues introduced through $\boldsymbol{Y}_{tr}$.

In contrast, with the label trick, we can equip C&S with trainable weights to further boost performance. To this end, we first split the training dataset into $\mathcal{D}_{in}$ and $\mathcal{D}_{out}$ as before. Then we multiply $\tilde{\boldsymbol{E}}_{in} = \gamma(\boldsymbol{P}_c(\boldsymbol{Y}_{in} - \tilde{\boldsymbol{Y}}_{in}))$, the correction matrix with respect to $\boldsymbol{Y}_{in}$, with a weight matrix $\boldsymbol{W}_c$. We also multiply the result after smoothing with another weight matrix $\boldsymbol{W}_s$. Thus the predictor under this split is

$$f_{TC\&S}(\boldsymbol{Y}_{in}, \tilde{\boldsymbol{Y}}; \boldsymbol{W}) = \boldsymbol{P}_s(\boldsymbol{Y}_{in} + (\boldsymbol{M}_{te} + \boldsymbol{M}_{out})(\tilde{\boldsymbol{Y}} + \tilde{\boldsymbol{E}}_{in}\boldsymbol{W}_c))\boldsymbol{W}_s$$

$$= \boldsymbol{P}_s((\boldsymbol{Y}_{in} + (\boldsymbol{M}_{te} + \boldsymbol{M}_{out})\tilde{\boldsymbol{Y}})\boldsymbol{W}_s + \boldsymbol{P}_s(\boldsymbol{M}_{te} + \boldsymbol{M}_{out})\tilde{\boldsymbol{E}}_{in}\hat{\boldsymbol{W}}_c, \quad (14)$$

where $\boldsymbol{W} = \{\hat{\boldsymbol{W}}_c, \boldsymbol{W}_s\}$ and $\hat{\boldsymbol{W}}_c = \boldsymbol{W}_c \boldsymbol{W}_s$. The objective function for optimizing $\boldsymbol{W}$ is

$$\mathcal{L}(\boldsymbol{W}) = \underset{splits}{\mathbb{E}} \Big[ \sum_{i \in \mathcal{D}_{out}} \ell(\boldsymbol{y}_i, f_{TC\&S}[\boldsymbol{Y}_{in}, \tilde{\boldsymbol{Y}}; \boldsymbol{W}]_i) \Big]. \quad (15)$$

Since this objective resolves the label leakage issue, it allows end-to-end training of C&S with gradients passing through both the neural network layers for computing $\tilde{\boldsymbol{Y}}$ and the C&S steps. At times, however, this approach may have disadvantages, including potential overfitting problems or inefficiencies due to computationally expensive backpropagation. Consequently, an alternative option is to preserve the two-stage training. In this situation, the base prediction in the first stage is the same as the original algorithm; however, we can nonetheless still train the C&S module as a post-processing step, with parameters as in (14).

*Table 1.* Accuracy results (%) of label propagation and trainable label propagation.

| Method | Label Propagation | Trainable Label Propagation |
|---|---|---|
| Cora-full | 66.44 ± 0.93 | **67.40 ± 0.96** |
| Pubmed | 83.45 ± 0.63 | **83.52 ± 0.59** |
| ArXiv | 67.11 ± 0.00 | **68.42 ± 0.01** |
| Products | 74.24 ± 0.00 | **75.61 ± 0.21** |

*Table 2.* $R^2$ on node regression tasks with GBDT base model and regular/end-to-end C&S.

| Method | C&S | Trainable C&S |
|---|---|---|
| House | 0.797 ± 0.005 | **0.854 ± 0.005** |
| County | 0.625 ± 0.048 | **0.788 ± 0.015** |
| VK | 0.163 ± 0.011 | **0.172 ± 0.011** |
| Avazu | 0.405 ± 0.036 | **0.413 ± 0.033** |

## 5 EXPERIMENTS

As mentioned previously, the effectiveness of the label trick in improving GNN performance has already been demonstrated in prior work, and therefore, our goal here is not to repeat these efforts. Instead, in this section we will focus on conducting experiments that complement our analysis from Section 3 and showcase the broader application scenarios from Section 4.

The label trick can actually be implemented in two ways. The first is the one that randomly splits the training nodes, and the second is the simpler version introduced herein with the deterministic one-versus-all splitting strategy, which does not require any random sampling. To differentiate the two versions, we denote the stochastic label trick with random splits by **label trick (S)**, and the deterministic one by **label trick (D)**. The latter is an efficient way to approximate the former with a higher splitting probability $\alpha$, which is sometimes advantageous in cases where the training process is slowed down by high $\alpha$. Accordingly, we conduct experiments with both, thus supporting our analysis with comparisons involving the two versions.

We use four relatively large datasets for evaluation, namely Cora-full, Pubmed (Sen et al., 2008), ogbn-arxiv and ogbn-products (Hu et al., 2020). For Cora-full and Pubmed, we randomly split the nodes into training, validation, and test datasets with the ratio of 6:2:2 using different random seeds. For ogbn-arxiv and ogbn-products, we adopt the standard split from OGB (Hu et al., 2020). We report the average classification accuracy and standard deviation after 10 runs with different random seeds, and these are the results on the test dataset when not otherwise specified. See Appendix C for further implementation details.

### 5.1 TRAINABLE LABEL PROPAGATION

We first investigate the performance of applying the label trick to label propagation as introduced in (2) in the absence of features, and compare it with the original label propagation algorithm. Table 1 shows that the trainable weights applied to the label propagation algorithm can boost the performance consistently. Given the notable simplicity of label propagation, this represents a useful enhancement.

### 5.2 DETERMINISTIC LABEL TRICK APPLIED TO GNNS WITH LINEAR LAYERS

We also test the deterministic label trick by applying it to different GNN architectures involving linear propagation layers along with (13). Due to the considerable computational effort required to produce $P$ and $C$ with large propagation steps for larger graphs, we only conduct tests on the Cora-full, Pubmed and ogbn-arxiv datasets, where the results are presented in Table 3.

*Table 3.* Accuracy results (%) with/without label trick (D).

| Method | SGC | | SIGN | | TWIRLS | |
|---|---|---|---|---|---|---|
| label trick (D) | ✗ | ✓ | ✗ | ✓ | ✗ | ✓ |
| Cora-full | **65.87 ± 0.61** | 65.81 ± 0.69 | **68.54 ± 0.76** | 68.44 ± 0.88 | 70.36 ± 0.51 | **70.40 ± 0.71** |
| Pubmed | 85.02 ± 0.43 | **85.23 ± 0.57** | 87.94 ± 0.52 | **88.09 ± 0.59** | 89.81 ± 0.56 | **90.08 ± 0.52** |
| ArXiv | 69.07 ± 0.01 | **70.22 ± 0.03** | 69.97 ± 0.16 | **70.98 ± 0.21** | 72.93 ± 0.19 | **73.22 ± 0.10** |

From these results we observe that on Pubmed and ogbn-arxiv, the deterministic label trick boosts the performance consistently on different models, while on Cora-full, it performs comparably. This is reasonable given that the training accuracy on Cora-full (not shown) is close to 100%, in which case the model does not benefit significantly from the ground-truth training labels, as the label information is already adequately embedded in the model.

### 5.3 EFFECT OF SPLITTING PROBABILITY

In terms of the effect of the splitting probability $\alpha$, we compare the accuracy results of a linear model and a three-layer GCN on ogbn-arxiv as shown in Figure 1. For the linear model, $\alpha$ serves as a

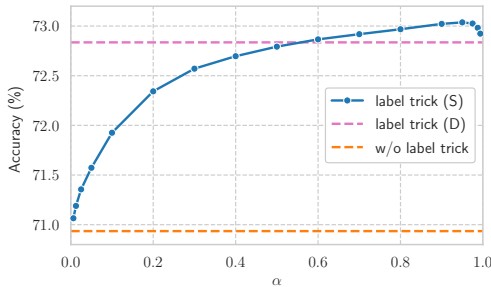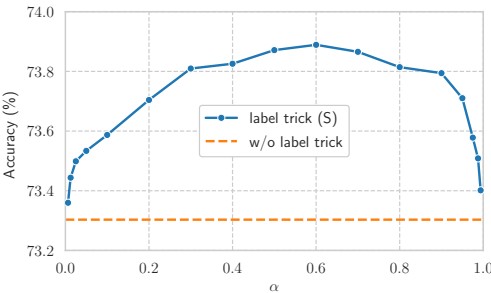

*Figure 1.* Validation accuracy varying $\alpha$. *Left*: linear propagation of features & labels, *Right*: GCN.

regularization coefficient. More specifically, when $\alpha$ tends to zero, the model converges to the one without the label trick, while when $\alpha$ tends to one, it converges to the case with the self-excluded label trick. For the the nonlinear GCN, $\alpha$ has a similar effect as predicted by theory for the linear models. As $\alpha$ decreases, the model converges to that without the label trick. Moreover, a larger $\alpha$ is preferable for linear models that may not require strong regularization like more complex GNNs.

### 5.4 Trainable Correct and Smooth

We also verify the effectiveness of our approach when applied to Correct and Smooth (C&S), as described in Section 4.3. Due to the significant impact of end-to-end training on the model, it is not suitable for direct comparison with vanilla C&S for a natural ablation. Therefore, in this experiment, we train the C&S as post-processing steps. In Table 4, we report the test and validation accuracy, showing that our method outperforms the vanilla C&S on Cora-full and Pubmed. And for ogbn-arxiv and ogbn-products, the trainable C&S performs better in terms of validation accuracy while is comparable to vanilla C&S in terms of test accuracy.

In principle, C&S can be applied to any base predictor model. To accommodate tabular node features, we choose to use gradient boosted decision trees (GBDT), which can be trained end-to-end using methods such as Chen et al. (2022); Ivanov & Prokhorenkova (2021) combined with the label trick to avoid data leakage issues as we have discussed. For evaluation, we adopt the four tabular node regression data sets from Ivanov & Prokhorenkova (2021) and train using the approach from Chen et al. (2022). Results are shown in Table 2, where the label trick can significantly improve performance.

*Table 4.* Test and validation accuracy (%) of C&S and trainable C&S with MLP as base predictor. Validation accuracy reported in parentheses.

| Method | MLP | MLP+C&S | MLP+Trainable C&S |
|---|---|---|---|
| Cora-full | 60.12 ± 0.29 (61.09 ± 0.39) | 66.95 ± 1.46 (68.26 ± 1.24) | **67.89 ± 1.37 (69.09 ± 1.25)** |
| Pubmed | 88.72 ± 0.34 (89.25 ± 0.26) | 89.12 ± 0.27 (89.45 ± 0.17) | **89.76 ± 0.17 (89.62 ± 0.18)** |
| ArXiv | 71.48 ± 0.15 (72.95 ± 0.05) | **73.05 ± 0.35** (74.01 ± 0.17) | 73.03 ± 0.18 (**74.44 ± 0.08**) |
| Products | 67.60 ± 0.15 (87.07 ± 0.05) | **83.16 ± 0.13** (91.70 ± 0.06) | 83.10 ± 0.15 (**91.99 ± 0.07**) |

## 6 Conclusion

In this work we closely examine from a theoretical prospective our recently proposed label trick, which enables the parallel propagation of labels and features and benefits various SOTA GNN architectures, and yet has thus far not be subject to rigorous analysis. In filling up this gap, we first introduce a deterministic self-excluded simplification of the label trick, and then prove that the full stochastic version can be regarded as introducing a regularization effect on the self-excluded label weights. Beyond this, we also discuss broader applications of the label trick with respect to: (i) Facilitating the introduction of trainable weights within graph-based methods that were previously either parameter-free (e.g., label propagation) or not end-to-end (e.g., C&S), and (ii) Eliminating the effects of randomness by incorporating self-excluded propagation within GNNs composed of linear propagation layers. We verify these applications and evaluate the performance gains against existing approaches over multiple benchmark datasets.

**Acknowledgments.** The Shanghai Jiao Tong University Team is supported by Shanghai Municipal Science and Technology Major Project (2021SHZDZX0102) and National Natural Science Foundation of China (62076161, 62177033). We would also like to thank Wu Wen Jun Honorary Doctoral Scholarship from AI Institute, Shanghai Jiao Tong University.

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

# A  ADDITIONAL EXPERIMENTS

## A.1  EFFECT OF TRAINING ACCURACY

Intuitively, the label trick introduces more information about the ground-truth labels in the training set to the model. In this experiment, we investigate the role of training accuracy in training with the label trick. We produce two new datasets by projecting the input features on Cora-full and Pubmed to 32 dimensions (i.e., Cora-full-32d and Pubmed-32d, respectively) via PCA. This is a much more challenging task because less feature information is available for prediction. We train SGC, SIGN and TWIRLS under the same settings as in Table 3. The following tables show the test and training accuracy on these datasets respectively, as well as results using original features as in Table 3. For each comparison, we use the same set of model hyperparameters with/without label trick, based on the best performance without label trick.

*Table 5.* Test accuracy (%) with/without label trick (D).

| Method | SGC | | SIGN | | TWIRLS | |
|---|---|---|---|---|---|---|
| label trick (D) | ✗ | ✓ | ✗ | ✓ | ✗ | ✓ |
| Cora-full-32d | 55.63 ± 0.53 | **62.37 ± 0.87** | 55.10 ± 0.55 | **62.05 ± 0.47** | 58.57 ± 0.48 | **61.74 ± 0.46** |
| Pubmed-32d | 83.15 ± 0.41 | **84.33 ± 0.68** | 84.75 ± 0.37 | **85.72 ± 0.84** | 84.94 ± 0.37 | **87.60 ± 0.44** |
| Cora-full | **65.87 ± 0.61** | 65.81 ± 0.69 | **68.54 ± 0.76** | 68.44 ± 0.88 | 70.36 ± 0.51 | **70.40 ± 0.71** |
| Pubmed | 85.02 ± 0.43 | **85.23 ± 0.57** | 87.94 ± 0.52 | **88.09 ± 0.59** | 89.81 ± 0.56 | **90.08 ± 0.52** |
| ArXiv | 69.07 ± 0.01 | **70.22 ± 0.03** | 69.97 ± 0.16 | **70.98 ± 0.21** | 72.93 ± 0.19 | **73.22 ± 0.10** |

*Table 6.* Training accuracy (%) with/without label trick (D). Each comparison shares the model hyperparameters.

| Method | SGC | | SIGN | | TWIRLS | |
|---|---|---|---|---|---|---|
| label trick (D) | ✗ | ✓ | ✗ | ✓ | ✗ | ✓ |
| Cora-full-32d | 57.69 ± 0.21 | **65.75 ± 0.19** | 57.20 ± 0.21 | **65.88 ± 0.27** | 63.44 ± 0.41 | **67.40 ± 0.36** |
| Pubmed-32d | 83.43 ± 0.35 | **84.74 ± 0.17** | 84.96 ± 0.13 | **86.05 ± 0.19** | 85.97 ± 0.22 | **88.43 ± 0.26** |
| Cora-full | 95.31 ± 0.16 | **97.59 ± 0.12** | 99.78 ± 0.03 | **99.99 ± 0.01** | **99.95 ± 0.01** | 99.76 ± 0.05 |
| Pubmed | 85.63 ± 0.17 | **85.69 ± 0.23** | 91.26 ± 0.12 | **91.43 ± 0.17** | 97.32 ± 0.30 | **97.48 ± 0.28** |
| ArXiv | 71.66 ± 0.01 | **74.94 ± 0.01** | 71.26 ± 0.08 | **74.94 ± 0.08** | 80.92 ± 0.17 | **85.08 ± 0.12** |

From Tables 5 and 6, we can see that the degree to which the label trick can improve accuracy is directly related to the training accuracy. When the training accuracy is near 100%, the label trick minimally benefits the model as the label information has already been adequately embedded in the model. However, when the training accuracy is lower, the label trick can increase both the training and test accuracy significantly (once the variance from the dataset divisions is taken into account, please see Appendix A.2).

Moreover, for large-scale graphs it may be less likely that high accuracy or overfitting is experienced during training due to the generous number of available samples. So in such cases, the label trick is more likely to be effective.

## A.2  EFFECT OF RANDOM DATASET DIVISION

Due to the random division of the dataset into training, validation and test dataset for Cora-full, Pubmed, and the regression datasets, the standard deviations across trials are considerably higher than when using fixed splits. Consequently, for many of the results, the performance gap remains consistent across trials such that the improvement is actually still significant.

As an illustration, Table 7 presents the itemized, trial-to-trial results that were averaged to produce the Cora-full results presented earlier in Table 1. This reveals that the label trick consistently improves performance over different dataset splits even after careful tuning of the hyperparameters for the baseline model. This phenomena is similarly present in many of other experimental results.

*Table 7.* Trail-to-trail accuracy results (%) on Cora-full.

| Trail | Label Propagation | Trainable Label Propagation |
|:---:|:---:|:---:|
| 1 | 65.52 | **67.57** |
| 2 | 65.77 | **66.03** |
| 3 | 64.84 | **66.28** |
| 4 | 66.00 | **66.23** |
| 5 | 68.10 | **68.58** |
| 6 | 67.39 | **67.87** |
| 7 | 67.29 | **68.07** |
| 8 | 66.86 | **68.83** |
| 9 | 66.48 | **67.85** |
| 10 | 66.13 | **66.73** |
| Overall | 66.44 ± 0.93 | **67.40 ± 0.96** |

### A.3 EFFECT OF REGULARIZATION FACTOR WITH SELF-EXCLUDED LINEAR PREDICTORS

In order to demonstrate the effect of the theoretically-derived regularization factor in Theorem 1 and Corollary 1, we apply the regularizer to classification experiments involving the deterministic self-excluded linear predictors from Section 4.2. Figure 2 provides such an example using the Cora dataset and the TWIRLS linear predictor plus regularization as is varied. Note that if, there is infinite penalty on the label weights which is equivalent to no labels; in contrast if the regularization factor is zero.

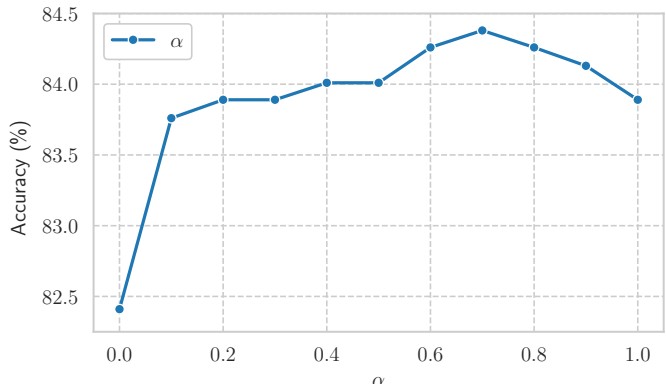

*Figure 2.* Accuracy results varying regularization coefficient with corresponding $\alpha$. When $\alpha = 0$, there is no actual label in use.

From Figure 2 we observe that including labels plus the theoretically-motivated regularization (with the right value for $\alpha$) can indeed improve performance. This suggests that even with a classification loss (which only loosely aligns with the stated theorem/corollary assumption of an MSE loss), the theory has some practical/predictive value in producing a useful regularization factor.

## B PROOFS

### B.1 PROOF OF THEOREM 1

We first define the mask matrix for input labels:

$$\boldsymbol{M}_{in} = \begin{pmatrix} \text{diag}(r_i) & \boldsymbol{0} \\ \boldsymbol{0} & \boldsymbol{0} \end{pmatrix}, r_i \sim \text{Bernoulli}(\alpha), \qquad (16)$$

where $\alpha \in (0, 1)$ is the label rate, and the mask matrix for output is defined as $\boldsymbol{M}_{out} = \boldsymbol{M}_{tr} - \boldsymbol{M}_{in}$. Then the objective is

$$\mathcal{L}(\boldsymbol{W}) = \mathop{\mathbb{E}}_{splits}[\|\boldsymbol{Y}_{out} - \boldsymbol{M}_{out}\boldsymbol{P}\tilde{\boldsymbol{Y}}_{in}\boldsymbol{W}\|_F^2]$$

$$= \mathop{\mathbb{E}}_{splits}\left[\left\|\boldsymbol{Y}_{out} - \frac{1}{\alpha}\boldsymbol{M}_{out}\boldsymbol{P}\boldsymbol{Y}_{in}\boldsymbol{W}\right\|_F^2\right]$$

$$= \mathop{\mathbb{E}}_{splits}[\|\boldsymbol{Y}_{out}\|_F^2] - \frac{2}{\alpha}\mathop{\mathbb{E}}_{splits}[\mathrm{tr}[\boldsymbol{Y}_{out}^T\boldsymbol{M}_{out}\boldsymbol{P}\boldsymbol{Y}_{in}\boldsymbol{W}]] + \frac{1}{\alpha^2}\mathop{\mathbb{E}}_{splits}[\|\boldsymbol{M}_{out}\boldsymbol{P}\boldsymbol{Y}_{in}\boldsymbol{W}\|_F^2]$$

$$= \mathop{\mathbb{E}}_{splits}[\|\boldsymbol{M}_{out}\boldsymbol{Y}_{tr}\|_F^2] - \frac{2}{\alpha}\mathop{\mathbb{E}}_{splits}[\mathrm{tr}[\boldsymbol{Y}_{tr}^T\boldsymbol{M}_{out}\boldsymbol{P}\boldsymbol{M}_{in}\boldsymbol{Y}_{tr}\boldsymbol{W}]]$$

$$+ \frac{1}{\alpha^2}\mathop{\mathbb{E}}_{splits}[\|\boldsymbol{M}_{out}\boldsymbol{P}\boldsymbol{M}_{in}\boldsymbol{Y}_{tr}\boldsymbol{W}\|_F^2].$$

$$(17)$$

Next we compute each term separately. For notational convenience, we denote the $i$-th row and $j$-th column entry of $\boldsymbol{M}_{in}$ by $M_{ij}$.

1. For $\mathbb{E}_{splits}[\|\boldsymbol{M}_{out}\boldsymbol{Y}_{tr}\|_F^2]$, we have

$$\mathop{\mathbb{E}}_{splits}[\|\boldsymbol{M}_{out}\boldsymbol{Y}_{tr}\|_F^2] = \mathop{\mathbb{E}}_{splits}[\mathrm{tr}[\boldsymbol{Y}_{tr}^T\boldsymbol{M}_{out}^T\boldsymbol{M}_{out}\boldsymbol{Y}_{tr}]]$$

$$= \mathop{\mathbb{E}}_{splits}[\mathrm{tr}[\boldsymbol{Y}_{tr}^T\boldsymbol{M}_{out}\boldsymbol{Y}_{tr}]]$$

$$= \mathop{\mathbb{E}}_{splits}[\mathrm{tr}[\boldsymbol{M}_{out}\boldsymbol{Y}_{tr}\boldsymbol{Y}_{tr}^T]]$$

$$= \mathrm{tr}[\mathop{\mathbb{E}}_{splits}[\boldsymbol{M}_{out}]\boldsymbol{Y}_{tr}\boldsymbol{Y}_{tr}^T]$$

$$= (1-\alpha)\,\mathrm{tr}[\boldsymbol{M}_{tr}\boldsymbol{Y}_{tr}\boldsymbol{Y}_{tr}^T]$$

$$= (1-\alpha)\,\mathrm{tr}[\boldsymbol{Y}_{tr}\boldsymbol{Y}_{tr}^T]$$

$$= (1-\alpha)\|\boldsymbol{Y}_{tr}\|_F^2.$$

$$(18)$$

2. For $\mathbb{E}_{splits}[\mathrm{tr}[\boldsymbol{Y}_{tr}^T\boldsymbol{M}_{out}\boldsymbol{P}\boldsymbol{M}_{in}\boldsymbol{Y}_{tr}\boldsymbol{W}]]$, since $\boldsymbol{M}_{out} = \boldsymbol{M}_{tr} - \boldsymbol{M}_{in}$, we then have

$$\mathop{\mathbb{E}}_{splits}[\mathrm{tr}[\boldsymbol{Y}_{tr}^T\boldsymbol{M}_{out}\boldsymbol{P}\boldsymbol{M}_{in}\boldsymbol{Y}_{tr}\boldsymbol{W}]]$$

$$= \mathop{\mathbb{E}}_{splits}[\mathrm{tr}[\boldsymbol{M}_{out}\boldsymbol{P}\boldsymbol{M}_{in}\boldsymbol{Y}_{tr}\boldsymbol{W}\boldsymbol{Y}_{tr}^T]]$$

$$= \mathrm{tr}[\mathop{\mathbb{E}}_{splits}[\boldsymbol{M}_{out}\boldsymbol{P}\boldsymbol{M}_{in}]\boldsymbol{Y}_{tr}\boldsymbol{W}\boldsymbol{Y}_{tr}^T]$$

$$= \mathrm{tr}[\mathop{\mathbb{E}}_{splits}[(\boldsymbol{M}_{tr} - \boldsymbol{M}_{in})\boldsymbol{P}\boldsymbol{M}_{in}]\boldsymbol{Y}_{tr}\boldsymbol{W}\boldsymbol{Y}_{tr}^T]$$

$$= \mathrm{tr}[(\boldsymbol{M}_{tr}\boldsymbol{P}\mathop{\mathbb{E}}_{splits}[\boldsymbol{M}_{in}] - \mathop{\mathbb{E}}_{splits}[\boldsymbol{M}_{in}\boldsymbol{P}\boldsymbol{M}_{in}])\boldsymbol{Y}_{tr}\boldsymbol{W}\boldsymbol{Y}_{tr}^T]$$

$$= \mathrm{tr}[(\alpha\boldsymbol{M}_{tr}\boldsymbol{P}\boldsymbol{M}_{tr} - \mathop{\mathbb{E}}_{splits}[\boldsymbol{M}_{in}\boldsymbol{P}\boldsymbol{M}_{in}])\boldsymbol{Y}_{tr}\boldsymbol{W}\boldsymbol{Y}_{tr}^T]$$

$$= \alpha\,\mathrm{tr}[\boldsymbol{M}_{tr}\boldsymbol{P}\boldsymbol{M}_{tr}\boldsymbol{Y}_{tr}\boldsymbol{W}\boldsymbol{Y}_{tr}^T] - \mathrm{tr}\left[\mathop{\mathbb{E}}_{splits}[\boldsymbol{M}_{in}\boldsymbol{P}\boldsymbol{M}_{in}]\boldsymbol{Y}_{tr}\boldsymbol{W}\boldsymbol{Y}_{tr}^T\right]$$

$$= \alpha\,\mathrm{tr}[\boldsymbol{Y}_{tr}^T\boldsymbol{P}\boldsymbol{Y}_{tr}\boldsymbol{W}] - \mathrm{tr}\left[\mathop{\mathbb{E}}_{splits}[\boldsymbol{M}_{in}\boldsymbol{P}\boldsymbol{M}_{in}]\boldsymbol{Y}_{tr}\boldsymbol{W}\boldsymbol{Y}_{tr}^T\right]$$

$$(19)$$

Let us calculate $\mathbb{E}_{splits}[\boldsymbol{M}_{in}\boldsymbol{P}\boldsymbol{M}_{in}]$. Since $\boldsymbol{M}_{in}$ is a diagonal matrix, we have

$$\mathop{\mathbb{E}}_{splits}[\boldsymbol{M}_{in}\boldsymbol{P}\boldsymbol{M}_{in}] = (\mathop{\mathbb{E}}_{splits}[M_{ii}M_{jj}P_{ij}]) = \alpha^2\boldsymbol{P}_{tr} + (\alpha - \alpha^2)\boldsymbol{C}_{tr}, \qquad (20)$$

where $\boldsymbol{P}_{tr} = \boldsymbol{M}_{tr}\boldsymbol{P}\boldsymbol{M}_{tr}$ and $\boldsymbol{C}_{tr} = \mathrm{diag}(\boldsymbol{P}_{tr})$. Then we have

$$\mathrm{tr}[\mathop{\mathbb{E}}_{splits}[\boldsymbol{M}_{in}\boldsymbol{P}\boldsymbol{M}_{in}]\boldsymbol{Y}_{tr}\boldsymbol{W}\boldsymbol{Y}_{tr}^T] = \alpha^2\,\mathrm{tr}[\boldsymbol{Y}_{tr}^T\boldsymbol{P}_{tr}\boldsymbol{Y}_{tr}\boldsymbol{W}] + (\alpha - \alpha^2)\,\mathrm{tr}[\boldsymbol{Y}_{tr}^T\boldsymbol{C}_{tr}\boldsymbol{Y}_{tr}\boldsymbol{W}].$$

$$(21)$$

Therefore,

$$
\begin{aligned}
\mathop{\mathbb{E}}_{splits} & [\operatorname{tr}[\boldsymbol{Y}_{tr}^T \boldsymbol{M}_{out} \boldsymbol{P} \boldsymbol{M}_{in} \boldsymbol{Y}_{tr} \boldsymbol{W}]] \\
& = (\alpha - \alpha^2) \operatorname{tr}[\boldsymbol{Y}_{tr}^T \boldsymbol{P}_{tr} \boldsymbol{Y}_{tr} \boldsymbol{W}] - (\alpha - \alpha^2) \operatorname{tr}[\boldsymbol{Y}_{tr}^T \boldsymbol{C}_{tr} \boldsymbol{Y}_{tr} \boldsymbol{W}] \\
& = (\alpha - \alpha^2) \operatorname{tr}[\boldsymbol{Y}_{tr}^T (\boldsymbol{P}_{tr} - \boldsymbol{C}_{tr}) \boldsymbol{Y}_{tr} \boldsymbol{W}].
\end{aligned} \tag{22}
$$

3. For $\mathbb{E}_{splits}[\|\boldsymbol{M}_{out} \boldsymbol{P} \boldsymbol{M}_{in} \boldsymbol{Y}_{tr} \boldsymbol{W}\|_F^2]$, we have

$$
\begin{aligned}
\mathop{\mathbb{E}}_{splits} & [\|\boldsymbol{M}_{out} \boldsymbol{P} \boldsymbol{M}_{in} \boldsymbol{Y}_{tr} \boldsymbol{W}\|_F^2] \\
& = \mathop{\mathbb{E}}_{splits} [\operatorname{tr}[\boldsymbol{W}^T \boldsymbol{Y}_{tr}^T \boldsymbol{M}_{in}^T \boldsymbol{P}^T \boldsymbol{M}_{out}^T \boldsymbol{M}_{out} \boldsymbol{P} \boldsymbol{M}_{in} \boldsymbol{Y}_{tr} \boldsymbol{W}]] \\
& = \mathop{\mathbb{E}}_{splits} [\operatorname{tr}[\boldsymbol{W}^T \boldsymbol{Y}_{tr}^T \boldsymbol{M}_{in}^T \boldsymbol{P}^T \boldsymbol{M}_{out} \boldsymbol{P} \boldsymbol{M}_{in} \boldsymbol{Y}_{tr} \boldsymbol{W}]] \\
& = \operatorname{tr}[\mathop{\mathbb{E}}_{splits} [\boldsymbol{W}^T \boldsymbol{Y}_{tr}^T \boldsymbol{M}_{in}^T \boldsymbol{P}^T (\boldsymbol{M}_{tr} - \boldsymbol{M}_{in}) \boldsymbol{P} \boldsymbol{M}_{in}] \boldsymbol{Y}_{tr} \boldsymbol{W}] \\
& = \operatorname{tr}[\mathop{\mathbb{E}}_{splits} [\boldsymbol{W}^T \boldsymbol{Y}_{tr}^T \boldsymbol{M}_{in}^T \boldsymbol{P}^T \boldsymbol{M}_{tr} \boldsymbol{P} \boldsymbol{M}_{in}] \boldsymbol{Y}_{tr} \boldsymbol{W}] \\
& \quad - \operatorname{tr}[\mathop{\mathbb{E}}_{splits} [\boldsymbol{W}^T \boldsymbol{Y}_{tr}^T \boldsymbol{M}_{in}^T \boldsymbol{P}^T \boldsymbol{M}_{in} \boldsymbol{P} \boldsymbol{M}_{in}] \boldsymbol{Y}_{tr} \boldsymbol{W}] \\
& = \operatorname{tr}[\boldsymbol{W}^T \boldsymbol{Y}_{tr}^T \mathop{\mathbb{E}}_{splits} [\boldsymbol{M}_{in}^T \boldsymbol{P}^T \boldsymbol{M}_{tr} \boldsymbol{P} \boldsymbol{M}_{in}] \boldsymbol{Y}_{tr} \boldsymbol{W}] \\
& \quad - \operatorname{tr}[\boldsymbol{W}^T \boldsymbol{Y}_{tr}^T \mathop{\mathbb{E}}_{splits} [\boldsymbol{M}_{in}^T \boldsymbol{P}^T \boldsymbol{M}_{in} \boldsymbol{P} \boldsymbol{M}_{in}] \boldsymbol{Y}_{tr} \boldsymbol{W}].
\end{aligned} \tag{23}
$$

Let us consider each term separately.

(a) To compute $\operatorname{tr}[\boldsymbol{W}^T \boldsymbol{Y}_{tr}^T \mathbb{E}_{splits}[\boldsymbol{M}_{in}^T \boldsymbol{P}^T \boldsymbol{M}_{tr} \boldsymbol{P} \boldsymbol{M}_{in}] \boldsymbol{Y}_{tr} \boldsymbol{W}]$, we first define $\boldsymbol{Q}_{tr} = \operatorname{diag}(\boldsymbol{P}_{tr}^2)^{\frac{1}{2}}$, and then

$$
\begin{aligned}
\mathop{\mathbb{E}}_{splits} \left[ \boldsymbol{M}_{in}^T \boldsymbol{P}^T \boldsymbol{M}_{tr} \boldsymbol{P} \boldsymbol{M}_{in} \right]_{ij} & = \mathop{\mathbb{E}}_{splits} \left[ \sum_{k=1}^m M_{ii} M_{jj} P_{ki} P_{kj} \right] \\
& = \mathop{\mathbb{E}}_{splits} [M_{ii} M_{jj}] \sum_{k=1}^m P_{ki} P_{kj} \\
& = \left[ \alpha^2 \boldsymbol{P}_{tr}^T \boldsymbol{P}_{tr} + (\alpha - \alpha^2) \operatorname{diag}(\boldsymbol{P}_{tr}^2) \right]_{ij} \\
& = \left[ \alpha^2 \boldsymbol{P}_{tr}^T \boldsymbol{P}_{tr} + (\alpha - \alpha^2)(\boldsymbol{Q}_{tr}^2) \right]_{ij}.
\end{aligned} \tag{24}
$$

Therefore,

$$
\begin{aligned}
\operatorname{tr}[\boldsymbol{W}^T \boldsymbol{Y}_{tr}^T & \mathop{\mathbb{E}}_{splits} [\boldsymbol{M}_{in}^T \boldsymbol{P}^T \boldsymbol{M}_{tr} \boldsymbol{P} \boldsymbol{M}_{in}] \boldsymbol{Y}_{tr} \boldsymbol{W}] \\
& = \alpha^2 \|\boldsymbol{P}_{tr} \boldsymbol{Y}_{tr} \boldsymbol{W}\|_F^2 + (\alpha - \alpha^2) \|\boldsymbol{Q}_{tr} \boldsymbol{Y}_{tr} \boldsymbol{W}\|_F^2.
\end{aligned} \tag{25}
$$

(b) To compute $\operatorname{tr}[\boldsymbol{W}^T \boldsymbol{Y}_{tr}^T \mathbb{E}_{splits}[\boldsymbol{M}_{in}^T \boldsymbol{P}^T \boldsymbol{M}_{in} \boldsymbol{P} \boldsymbol{M}_{in}] \boldsymbol{Y}_{tr} \boldsymbol{W}]$, we have that

$$
\begin{aligned}
\mathop{\mathbb{E}}_{splits} \left[ \boldsymbol{M}_{in} \boldsymbol{P}^T \boldsymbol{M}_{in} \boldsymbol{P} \boldsymbol{M}_{in} \right]_{ij} & = \mathop{\mathbb{E}}_{splits} [M_{ii} M_{jj} \sum_{k=1}^n M_{kk} P_{ki} P_{kj}] \\
& = \sum_{k=1}^n \mathop{\mathbb{E}}_{splits} [M_{ii} M_{jj} M_{kk}] P_{ki} P_{kj}.
\end{aligned} \tag{26}
$$

• For the diagonal entries, where $i = j$, we have

$$
\sum_{k=1}^n \mathop{\mathbb{E}}_{splits} [M_{ii} M_{jj} M_{kk}] P_{ki} P_{kj} = \sum_{k=1}^n \alpha^2 P_{tr,ki}^2 + (\alpha - \alpha^2) P_{tr,ii}^2, \tag{27}
$$

and therefore,

$$
\begin{aligned}
\operatorname{diag} \left( \mathop{\mathbb{E}}_{splits} [\boldsymbol{M}_{in} \boldsymbol{P}^T \boldsymbol{M}_{in} \boldsymbol{P} \boldsymbol{M}_{in}] \right) & = \alpha^2 \operatorname{diag}(\boldsymbol{P}_{tr}^2) + (\alpha - \alpha^2) \boldsymbol{C}_{tr}^2 \\
& = \alpha^2 \boldsymbol{Q}_{tr}^2 + (\alpha - \alpha^2) \boldsymbol{C}_{tr}^2.
\end{aligned} \tag{28}
$$

- For the off-diagonal entries, where $i \neq j$, we have

$$\sum_k \mathop{\mathbb{E}}_{splits}[M_{ii}M_{jj}M_{kk}]P_{ki}P_{kj} = \alpha^3 \sum_{k \neq i \wedge k \neq j} P_{tr,ki}P_{tr,kj} + \alpha^2(P_{tr,ii}P_{tr,ij} + P_{tr,ji}P_{tr,jj}),$$

(29)

and

$$\mathop{\mathbb{E}}_{splits}[\boldsymbol{M}_{in}\boldsymbol{P}^T\boldsymbol{M}_{in}\boldsymbol{P}\boldsymbol{M}_{in}]_{ij}$$

$$= \left[\alpha^3 \boldsymbol{P}_{tr}^2 + (\alpha^2 - \alpha^3)(\boldsymbol{C}_{tr}\boldsymbol{P}_{tr} + \boldsymbol{P}_{tr}\boldsymbol{C}_{tr})\right]_{ij}.$$

(30)

From (28) and (30), we know that

$$\mathop{\mathbb{E}}_{splits}[\boldsymbol{M}_{in}\boldsymbol{P}^T\boldsymbol{M}_{in}\boldsymbol{P}\boldsymbol{M}_{in}]$$

$$= \alpha^3 \boldsymbol{P}_{tr}^2 + (\alpha^2 - \alpha^3)\boldsymbol{Q}_{tr}^2 + (\alpha - 3\alpha^2 + 2\alpha^3)\boldsymbol{C}_{tr}^2 + (\alpha^2 - \alpha^3)(\boldsymbol{C}_{tr}\boldsymbol{P}_{tr} + \boldsymbol{P}_{tr}\boldsymbol{C}_{tr}).$$

(31)

Unfortunately because $\boldsymbol{C}_{tr}\boldsymbol{P}_{tr} + \boldsymbol{P}_{tr}\boldsymbol{C}_{tr}$ is not necessarily positive semi-definite, we cannot simplify the proof using the Cholesky decomposition. So instead we proceed as follows:

$$\text{tr}[\boldsymbol{W}^T\boldsymbol{Y}_{tr}^T E[\boldsymbol{M}_{in}^T\boldsymbol{P}_{tr}\boldsymbol{M}_{in}\boldsymbol{P}_{tr}\boldsymbol{M}_{in}]\boldsymbol{Y}_{tr}\boldsymbol{W}]$$

$$= \text{tr}[\boldsymbol{W}^T\boldsymbol{Y}_{tr}^T(\alpha^3\boldsymbol{P}_{tr}^2 + (\alpha^2 - \alpha^3)\,\text{diag}(\boldsymbol{P}_{tr}^2) + (\alpha^2 - \alpha^3)(\boldsymbol{C}_{tr}\boldsymbol{P}_{tr} + \boldsymbol{P}_{tr}\boldsymbol{C}_{tr})$$

$$+ (\alpha^2 - \alpha^3)\,\text{diag}((\boldsymbol{C}_{tr}\boldsymbol{P}_{tr} + \boldsymbol{P}_{tr}\boldsymbol{C}_{tr})) + (\alpha - \alpha^2)\boldsymbol{C}_{tr}^2)\boldsymbol{Y}_{tr}\boldsymbol{W}]$$

$$= \text{tr}[\alpha^3\boldsymbol{W}^T\boldsymbol{Y}_{tr}^T\boldsymbol{P}_{tr}^2\boldsymbol{Y}_{tr}\boldsymbol{W} + (\alpha^2 - \alpha^3)\boldsymbol{W}^T\boldsymbol{Y}_{tr}^T\,\text{diag}(\boldsymbol{P}_{tr}^2)\boldsymbol{Y}_{tr}\boldsymbol{W}$$

$$+ (\alpha^2 - \alpha^3)\boldsymbol{W}^T\boldsymbol{Y}_{tr}^T(\boldsymbol{C}_{tr}\boldsymbol{P}_{tr} + \boldsymbol{P}_{tr}\boldsymbol{C}_{tr})\boldsymbol{Y}_{tr}\boldsymbol{W}$$

$$+ (\alpha - 3\alpha^2 + 2\alpha^3)\boldsymbol{W}^T\boldsymbol{Y}_{tr}^T\boldsymbol{C}_{tr}^2\boldsymbol{Y}_{tr}\boldsymbol{W}]$$

$$= \alpha^3\|\boldsymbol{P}_{tr}\boldsymbol{Y}_{tr}\boldsymbol{W}\|_F^2 + (\alpha^2 - \alpha^3)\|\boldsymbol{Q}_{tr}\boldsymbol{Y}_{tr}\boldsymbol{W}\|_F^2 + (\alpha - 3\alpha^2 + 2\alpha^3)\|\boldsymbol{C}_{tr}\boldsymbol{Y}_{tr}\boldsymbol{W}\|_F^2$$

$$+ (\alpha^2 - \alpha^3)\,\text{tr}[\boldsymbol{W}^T\boldsymbol{Y}_{tr}^T(\boldsymbol{C}_{tr}\boldsymbol{P}_{tr} + \boldsymbol{P}_{tr}\boldsymbol{C}_{tr})\boldsymbol{Y}_{tr}\boldsymbol{W}].$$

(32)

Combining the three terms, we get

$$\mathop{\mathbb{E}}_{splits}[\|\boldsymbol{M}_{out}\boldsymbol{P}\boldsymbol{M}_{in}\boldsymbol{Y}_{tr}\boldsymbol{W}\|_F^2]$$

$$= (\alpha^2 - \alpha^3)\|\boldsymbol{P}_{tr}\boldsymbol{Y}_{tr}\boldsymbol{W}\|_F^2 + (\alpha - 2\alpha^2 + \alpha^3)\|\boldsymbol{Q}_{tr}\boldsymbol{Y}_{tr}\boldsymbol{W}\|_F^2$$

$$- (\alpha^2 - \alpha^3)\,\text{tr}[\boldsymbol{W}^T\boldsymbol{Y}_{tr}(\boldsymbol{P}_{tr}\boldsymbol{C}_{tr} + \boldsymbol{C}_{tr}\boldsymbol{P}_{tr})\boldsymbol{Y}_{tr}\boldsymbol{W}] - (\alpha - 3\alpha^2 + 2\alpha^3)\|\boldsymbol{C}_{tr}\boldsymbol{Y}_{tr}\boldsymbol{W}\|_F^2.$$

(33)

The overall result $\mathcal{L}(\boldsymbol{W})$ is

$$\mathop{\mathbb{E}}_{splits}[\|\boldsymbol{Y}_{out} - \boldsymbol{M}_{out}\boldsymbol{P}\tilde{\boldsymbol{Y}}_{in}\boldsymbol{W}\|_F^2]$$

$$= \mathop{\mathbb{E}}_{splits}[\|\boldsymbol{M}_{out}\boldsymbol{Y}_{tr}\|_F^2] - \frac{2}{\alpha}\mathop{\mathbb{E}}_{splits}[\text{tr}[\boldsymbol{Y}_{tr}^T\boldsymbol{M}_{out}\boldsymbol{P}\boldsymbol{M}_{in}\boldsymbol{Y}_{tr}\boldsymbol{W}]]$$

$$+ \frac{1}{\alpha^2}\mathop{\mathbb{E}}_{splits}[\|\boldsymbol{M}_{out}\boldsymbol{P}\boldsymbol{M}_{in}\boldsymbol{Y}_{tr}\boldsymbol{W}\|_F^2]$$

$$= (1 - \alpha)\|\boldsymbol{Y}_{tr}\|_F^2 - 2(1 - \alpha)\,\text{tr}[\boldsymbol{Y}_{tr}^T(\boldsymbol{P}_{tr} - \boldsymbol{C}_{tr})\boldsymbol{Y}_{tr}\boldsymbol{W}] + (1 - \alpha)\|\boldsymbol{P}_{tr}\boldsymbol{Y}_{tr}\boldsymbol{W}\|_F^2$$

$$+ (\frac{1}{\alpha} - 2 + \alpha)\|\boldsymbol{Q}_{tr}\boldsymbol{Y}_{tr}\boldsymbol{W}\|_F^2 - (1 - \alpha)\,\text{tr}[\boldsymbol{W}^T\boldsymbol{Y}_{tr}(\boldsymbol{P}_{tr}\boldsymbol{C}_{tr} + \boldsymbol{C}_{tr}\boldsymbol{P}_{tr})\boldsymbol{Y}_{tr}\boldsymbol{W}]$$

$$- (\frac{1}{\alpha} - 3 + 2\alpha)\|\boldsymbol{C}_{tr}\boldsymbol{Y}_{tr}\boldsymbol{W}\|_F^2,$$

(34)

where $\boldsymbol{C}_{tr} = \mathrm{diag}(\boldsymbol{P}_{tr})$ and $\boldsymbol{Q}_{tr} = \mathrm{diag}(\boldsymbol{P}_{tr}^2)^{\frac{1}{2}}$.

Next we compute $\|\boldsymbol{Y}_{tr} - \boldsymbol{M}_{tr}(\boldsymbol{P}-\boldsymbol{C})\boldsymbol{Y}_{tr}\boldsymbol{W}\|_F^2$. Since

$$
\begin{aligned}
\|\boldsymbol{M}_{tr}(\boldsymbol{P}-\boldsymbol{C})\boldsymbol{Y}_{tr}\boldsymbol{W}\|_F^2 &= \|(\boldsymbol{P}_{tr}-\boldsymbol{C}_{tr})\boldsymbol{Y}_{tr}\boldsymbol{W}\|_F^2 \\
&= \mathrm{tr}[\boldsymbol{W}^T\boldsymbol{Y}_{tr}^T(\boldsymbol{P}_{tr}-\boldsymbol{C}_{tr})^T(\boldsymbol{P}_{tr}-\boldsymbol{C}_{tr})\boldsymbol{Y}_{tr}\boldsymbol{W}] \\
&= \|\boldsymbol{P}_{tr}\boldsymbol{Y}_{tr}\boldsymbol{W}\|_F^2 + \|\boldsymbol{C}_{tr}\boldsymbol{Y}_{tr}\boldsymbol{W}\|_F^2 - \mathrm{tr}[\boldsymbol{W}^T\boldsymbol{Y}_{tr}(\boldsymbol{P}_{tr}\boldsymbol{C}_{tr}+\boldsymbol{C}_{tr}\boldsymbol{P}_{tr})\boldsymbol{Y}_{tr}\boldsymbol{W}],
\end{aligned}
\tag{35}
$$

we have that

$$
\begin{aligned}
&\|\boldsymbol{Y}_{tr} - \boldsymbol{M}_{tr}(\boldsymbol{P}-\boldsymbol{C})\boldsymbol{Y}_{tr}\boldsymbol{W}\|_F^2 \\
&= \|\boldsymbol{Y}_{tr}\|_F^2 + \|\boldsymbol{P}_{tr}\boldsymbol{Y}_{tr}\boldsymbol{W}\|_F^2 + \|\boldsymbol{C}_{tr}\boldsymbol{Y}_{tr}\boldsymbol{W}\|_F^2 \\
&\quad - \mathrm{tr}[\boldsymbol{W}^T\boldsymbol{Y}_{tr}(\boldsymbol{P}_{tr}\boldsymbol{C}_{tr}+\boldsymbol{C}_{tr}\boldsymbol{P}_{tr})\boldsymbol{Y}_{tr}\boldsymbol{W}] - 2\,\mathrm{tr}[\boldsymbol{Y}_{tr}^T(\boldsymbol{P}_{tr}-\boldsymbol{C}_{tr})\boldsymbol{Y}_{tr}\boldsymbol{W}].
\end{aligned}
\tag{36}
$$

Therefore,

$$
\begin{aligned}
&\frac{1}{1-\alpha} \underset{splits}{\mathbb{E}}[\|\boldsymbol{Y}_{out} - \boldsymbol{M}_{out}\boldsymbol{P}\tilde{\boldsymbol{Y}}_{in}\boldsymbol{W}\|_F^2] \\
&= \|\boldsymbol{Y}_{tr}\|_F^2 - 2\,\mathrm{tr}[\boldsymbol{Y}_{tr}^T(\boldsymbol{P}_{tr}-\boldsymbol{C}_{tr})\boldsymbol{Y}_{tr}\boldsymbol{W}] + \|\boldsymbol{P}_{tr}\boldsymbol{Y}_{tr}\boldsymbol{W}\|_F^2 \\
&\quad + \frac{1-\alpha}{\alpha}\|\boldsymbol{Q}_{tr}\boldsymbol{Y}_{tr}\boldsymbol{W}\|_F^2 - \mathrm{tr}[\boldsymbol{W}^T\boldsymbol{Y}_{tr}(\boldsymbol{P}_{tr}\boldsymbol{C}_{tr}+\boldsymbol{C}_{tr}\boldsymbol{P}_{tr})\boldsymbol{Y}_{tr}\boldsymbol{W}] + \frac{2\alpha-1}{\alpha}\|\boldsymbol{C}_{tr}\boldsymbol{Y}_{tr}\boldsymbol{W}\|_F^2 \\
&= \|\boldsymbol{Y}_{tr} - \boldsymbol{M}_{tr}(\boldsymbol{P}-\boldsymbol{C})\boldsymbol{Y}_{tr}\boldsymbol{W}\|_F^2 + \frac{1-\alpha}{\alpha}\|\boldsymbol{Q}_{tr}\boldsymbol{Y}_{tr}\boldsymbol{W}\|_F^2 - \frac{1-\alpha}{\alpha}\|\boldsymbol{C}_{tr}\boldsymbol{Y}_{tr}\boldsymbol{W}\|_F^2.
\end{aligned}
\tag{37}
$$

Let $\boldsymbol{U} = (\mathrm{diag}(\boldsymbol{P}^T\boldsymbol{P}) - \boldsymbol{C}^T\boldsymbol{C})^{\frac{1}{2}}$, and $\boldsymbol{\Gamma} = \boldsymbol{U}\boldsymbol{Y}_{tr}$. Then we have

$$
\begin{aligned}
&\frac{1}{1-\alpha}\underset{splits}{\mathbb{E}}[\|\boldsymbol{Y}_{out} - \boldsymbol{M}_{out}\boldsymbol{P}\tilde{\boldsymbol{Y}}_{in}\boldsymbol{W}\|_F^2] \\
&\quad = \|\boldsymbol{Y}_{tr} - \boldsymbol{M}_{tr}(\boldsymbol{P}-\boldsymbol{C})\boldsymbol{Y}_{tr}\boldsymbol{W}\|_F^2 + \frac{1-\alpha}{\alpha}\|\boldsymbol{M}_{tr}\boldsymbol{U}\boldsymbol{Y}_{tr}\boldsymbol{W}\|_F^2 \\
&\quad = \|\boldsymbol{Y}_{tr} - \boldsymbol{M}_{tr}(\boldsymbol{P}-\boldsymbol{C})\boldsymbol{Y}_{tr}\boldsymbol{W}\|_F^2 + \frac{1-\alpha}{\alpha}\|\boldsymbol{U}\boldsymbol{Y}_{tr}\boldsymbol{W}\|_F^2 \\
&\quad = \|\boldsymbol{Y}_{tr} - \boldsymbol{M}_{tr}(\boldsymbol{P}-\boldsymbol{C})\boldsymbol{Y}_{tr}\boldsymbol{W}\|_F^2 + \frac{1-\alpha}{\alpha}\|\boldsymbol{\Gamma}\boldsymbol{W}\|_F^2.
\end{aligned}
\tag{38}
$$

## B.2 PROOF OF COROLLARY 1

We have that

$$
\begin{aligned}
&\frac{1}{1-\alpha}\underset{splits}{\mathbb{E}}\left[\|\boldsymbol{Y}_{out} - \boldsymbol{M}_{out}\boldsymbol{P}\boldsymbol{X}\boldsymbol{W}_x - \boldsymbol{M}_{out}\boldsymbol{P}\tilde{\boldsymbol{Y}}_{in}\boldsymbol{W}_y\|_F^2\right] \\
&= \frac{1}{1-\alpha}\underset{splits}{\mathbb{E}}\Big[\|\boldsymbol{Y}_{out} - \boldsymbol{M}_{out}\boldsymbol{P}\tilde{\boldsymbol{Y}}_{in}\boldsymbol{W}_y\|_F^2 \\
&\quad + \|\boldsymbol{M}_{out}\boldsymbol{P}\boldsymbol{X}\boldsymbol{W}_x\|_F^2 - 2\,\mathrm{tr}[\boldsymbol{Y}_{out}^T\boldsymbol{M}_{out}\boldsymbol{P}\boldsymbol{X}\boldsymbol{W}_x] + \frac{2}{\alpha}\mathrm{tr}[\boldsymbol{W}_y^T\boldsymbol{Y}_{in}^T\boldsymbol{P}^T\boldsymbol{M}_{out}\boldsymbol{P}\boldsymbol{X}\boldsymbol{W}_x]\Big] \\
&= \|\boldsymbol{Y}_{tr} - (\boldsymbol{P}_{tr}-\boldsymbol{C}_{tr})\boldsymbol{Y}_{tr}\boldsymbol{W}_y\|_F^2 + \frac{1-\alpha}{\alpha}\|\boldsymbol{\Gamma}\boldsymbol{W}_y\|_F^2 \\
&\quad + \|\boldsymbol{M}_{tr}\boldsymbol{P}\boldsymbol{X}\boldsymbol{W}_x\|_F^2 - 2\,\mathrm{tr}[\boldsymbol{Y}_{tr}^T\boldsymbol{P}\boldsymbol{X}\boldsymbol{W}_x] + 2\,\mathrm{tr}[\boldsymbol{W}_y^T\boldsymbol{Y}_{tr}^T(\boldsymbol{P}_{tr}-\boldsymbol{C}_{tr})\boldsymbol{P}\boldsymbol{X}\boldsymbol{W}_x] \\
&= \|\boldsymbol{Y}_{tr} - \boldsymbol{M}_{tr}\boldsymbol{P}\boldsymbol{X}\boldsymbol{W}_x - (\boldsymbol{P}_{tr}-\boldsymbol{C}_{tr})\boldsymbol{Y}_{tr}\boldsymbol{W}_y\|_F^2 + \frac{1-\alpha}{\alpha}\|\boldsymbol{\Gamma}\boldsymbol{W}_y\|_F^2 \\
&= \|\boldsymbol{Y}_{tr} - \boldsymbol{M}_{tr}\boldsymbol{P}\boldsymbol{X}\boldsymbol{W}_x - \boldsymbol{M}_{tr}(\boldsymbol{P}-\boldsymbol{C})\boldsymbol{Y}_{tr}\boldsymbol{W}_y\|_F^2 + \frac{1-\alpha}{\alpha}\|\boldsymbol{\Gamma}\boldsymbol{W}_y\|_F^2,
\end{aligned}
\tag{39}
$$

which proves the conclusion.

### B.3 PROOF OF THEOREM 2

We begin by noting that

$$
\begin{aligned}
\mathbb{E}_{splits}[\boldsymbol{M}_{out}\boldsymbol{P}\tilde{\boldsymbol{Y}}_{in}\boldsymbol{W}_y] &= \frac{1}{\alpha}\mathbb{E}_{splits}[\boldsymbol{M}_{tr}\boldsymbol{P}\boldsymbol{Y}_{in}\boldsymbol{W}_y] - \frac{1}{\alpha}\mathbb{E}_{splits}[\boldsymbol{M}_{in}\boldsymbol{P}\boldsymbol{Y}_{in}\boldsymbol{W}_y] \\
&= \frac{1}{\alpha}\boldsymbol{M}_{tr}\boldsymbol{P}\mathbb{E}_{splits}[\boldsymbol{M}_{in}]\boldsymbol{Y}_{tr}\boldsymbol{W}_y - \frac{1}{\alpha}\mathbb{E}_{splits}[\boldsymbol{M}_{in}\boldsymbol{P}\boldsymbol{M}_{in}]\boldsymbol{Y}_{tr}\boldsymbol{W}_y \\
&= \boldsymbol{M}_{tr}\boldsymbol{P}\boldsymbol{Y}_{tr}\boldsymbol{W}_y - (\alpha\boldsymbol{P}_{tr} - \alpha\boldsymbol{C}_{tr} + \boldsymbol{C}_{tr})\boldsymbol{Y}_{tr}\boldsymbol{W}_y \\
&= (1-\alpha)\boldsymbol{M}_{tr}\boldsymbol{P}\boldsymbol{Y}_{tr}\boldsymbol{W}_y - (1-\alpha)\boldsymbol{C}_{tr}\boldsymbol{Y}_{tr}\boldsymbol{W}_y \\
&= (1-\alpha)\boldsymbol{M}_{tr}(\boldsymbol{P} - \boldsymbol{C})\boldsymbol{Y}_{tr}\boldsymbol{W}_y.
\end{aligned}
\tag{40}
$$

And then by Jensen's inequality we have

$$
\begin{aligned}
&\mathbb{E}_{splits}[\text{CrossEntropy}_{\mathcal{D}_{out}}(\boldsymbol{Y}_{out}, \boldsymbol{M}_{out}\boldsymbol{P}\boldsymbol{X}\boldsymbol{W}_x + \boldsymbol{M}_{out}\boldsymbol{P}\tilde{\boldsymbol{Y}}_{in}\boldsymbol{W}_y)] \\
&\geq \text{CrossEntropy}_{\mathcal{D}_{out}}(\boldsymbol{Y}_{out}, \mathbb{E}_{splits}[\boldsymbol{M}_{out}\boldsymbol{P}\boldsymbol{X}\boldsymbol{W}_x + \boldsymbol{M}_{out}\boldsymbol{P}\tilde{\boldsymbol{Y}}_{in}\boldsymbol{W}_y]) \\
&= \text{CrossEntropy}_{\mathcal{D}_{tr}}(\boldsymbol{Y}_{tr}, (1-\alpha)(\boldsymbol{P}\boldsymbol{X}\boldsymbol{W}_x + (\boldsymbol{P} - \boldsymbol{C})\boldsymbol{Y}_{tr}\boldsymbol{W}_y))).
\end{aligned}
\tag{41}
$$

Notice based on the nature of cross-entropy, if $0 < \alpha < 1$, we have

$$
\text{CrossEntropy}(\boldsymbol{Z}, (1-\alpha)\tilde{\boldsymbol{Z}}) \geq (1-\alpha)\text{CrossEntropy}(\boldsymbol{Z}, \tilde{\boldsymbol{Z}}).
\tag{42}
$$

Therefore,

$$
\begin{aligned}
&\frac{1}{1-\alpha}\mathbb{E}_{splits}[\text{CrossEntropy}_{\mathcal{D}_{out}}(\boldsymbol{Y}_{out}, \boldsymbol{P}\boldsymbol{X}\boldsymbol{W}_x + \boldsymbol{P}\tilde{\boldsymbol{Y}}_{in}\boldsymbol{W}_y)] \\
&\geq \text{CrossEntropy}_{\mathcal{D}_{tr}}(\boldsymbol{Y}_{tr}, \boldsymbol{X}\boldsymbol{W}_x + (\boldsymbol{P} - \boldsymbol{C})\boldsymbol{Y}_{tr}\boldsymbol{W}_y).
\end{aligned}
\tag{43}
$$

### B.4 PROOF OF THEOREM 3

Although the label trick splits are drawn randomly using an iid Bernoulli distribution to sample the elements to be added to either $\mathcal{D}_{in}$ or $\mathcal{D}_{out}$, we can equivalently decompose this process into two parts. First, we draw $|\mathcal{D}_{out}|$ from a binomial distribution $P[|\mathcal{D}_{out}|] = \text{Binom}[m, 1-\alpha]$, where $m$ is the number of trials and $1-\alpha$ is the success probability. And then we choose the elements to be assigned to $\mathcal{D}_{out}$ uniformly over the possible combinations $\binom{m}{|\mathcal{D}_{out}|}$.

Given that when $|\mathcal{D}_{out}| = 0$ there is no contribution to the loss, we may therefore reexpress the lefthand side of (12), excluding the limit, as

$$
\begin{aligned}
&\frac{1}{1-\alpha}\mathbb{E}_{splits}\Big[\sum_{i\in\mathcal{D}_{out}}\ell\big(\boldsymbol{y}_i, f_{GNN}[\boldsymbol{X}, \boldsymbol{Y}_{in}; \boldsymbol{\mathcal{W}}]_i\big)\Big] = \\
&\qquad \frac{P[|\mathcal{D}_{out}| = 1]}{1-\alpha}\mathbb{E}_{P[\mathcal{D}_{out}||\mathcal{D}_{out}|=1]}\Big[\sum_{i\in\mathcal{D}_{out}}\ell\big(\boldsymbol{y}_i, f_{GNN}[\boldsymbol{X}, \boldsymbol{Y}_{in}; \boldsymbol{\mathcal{W}}]_i\big)\Big] \\
&\qquad + \frac{P[|\mathcal{D}_{out}| > 1]}{1-\alpha}\mathbb{E}_{P[\mathcal{D}_{out}||\mathcal{D}_{out}|>1]}\Big[\sum_{i\in\mathcal{D}_{out}}\ell\big(\boldsymbol{y}_i, f_{GNN}[\boldsymbol{X}, \boldsymbol{Y}_{in}; \boldsymbol{\mathcal{W}}]_i\big)\Big],
\end{aligned}
\tag{44}
$$

where it naturally follows that

$$
\mathbb{E}_{P[\mathcal{D}_{out}||\mathcal{D}_{out}|=1]}\Big[\sum_{i\in\mathcal{D}_{out}}\ell\big(\boldsymbol{y}_i, f_{GNN}[\boldsymbol{X}, \boldsymbol{Y}_{in}; \boldsymbol{\mathcal{W}}]_i\big)\Big] = \frac{1}{m}\sum_{i=1}^{m}\ell\big(\boldsymbol{y}_i, f_{GNN}[\boldsymbol{X}, \boldsymbol{Y}_{tr}-\boldsymbol{Y}_i; \boldsymbol{\mathcal{W}}]_i\big).
\tag{45}
$$

We also have that

$$
\frac{P[|\mathcal{D}_{out}| > 1]}{1-\alpha} = \frac{\sum_{i=2}^{m}\binom{m}{i}(1-\alpha)^i\alpha^{m-i}}{1-\alpha} = \sum_{i=2}^{m}\binom{m}{i}(1-\alpha)^{i-1}\alpha^{m-i}
\tag{46}
$$

and

$$\frac{P\left[|\mathcal{D}_{out}| = 1\right]}{1 - \alpha} = \frac{m(1 - \alpha)\alpha^{m-1}}{1 - \alpha} = m\alpha^{m-1}. \tag{47}$$

From these expressions, we then have that

$$\lim_{\alpha \to 1} \frac{P\left[|\mathcal{D}_{out}| > 1\right]}{1 - \alpha} = 0 \ \text{ and } \ \lim_{\alpha \to 1} \frac{P\left[|\mathcal{D}_{out}| = 1\right]}{1 - \alpha} = m. \tag{48}$$

And given the assumption that $\left|\sum_{i \in \mathcal{D}_{out}} \ell\big(\boldsymbol{y}_i, \ f_{GNN}[\boldsymbol{X}, \boldsymbol{Y}_{in}; \mathcal{W}]_i\big)\right| < B < \infty$ for some $B$, we also have

$$\mathop{\mathbb{E}}_{P\left[\mathcal{D}_{out}||\mathcal{D}_{out}|>1\right]} \Big[ \sum_{i \in \mathcal{D}_{out}} \ell\big(\boldsymbol{y}_i, \ f_{GNN}[\boldsymbol{X}, \boldsymbol{Y}_{in}; \mathcal{W}]_i\big) \Big] < \mathop{\mathbb{E}}_{P\left[\mathcal{D}_{out}||\mathcal{D}_{out}|>1\right]} \Big[ B \Big] = B, \tag{49}$$

where the bound is independent of $\alpha$. Consequently,

$$\lim_{\alpha \to 1} \frac{P\left[|\mathcal{D}_{out}| > 1\right]}{1 - \alpha} \mathop{\mathbb{E}}_{P\left[\mathcal{D}_{out}||\mathcal{D}_{out}|>1\right]} \Big[ \sum_{i \in \mathcal{D}_{out}} \ell\big(\boldsymbol{y}_i, \ f_{GNN}[\boldsymbol{X}, \boldsymbol{Y}_{in}; \mathcal{W}]_i\big) \Big] = 0 \tag{50}$$

such that when combined with the other expressions above, the conclusion

$$\lim_{\alpha \to 1} \left\{ \frac{1}{1 - \alpha} \mathop{\mathbb{E}}_{splits} \Big[ \sum_{i \in \mathcal{D}_{out}} \ell\big(\boldsymbol{y}_i, \ f_{GNN}[\boldsymbol{X}, \boldsymbol{Y}_{in}; \mathcal{W}]_i\big) \Big] \right\} = \sum_{i=1}^{m} \ell\big(\boldsymbol{y}_i, \ f_{GNN}[\boldsymbol{X}, \boldsymbol{Y}_{tr} - \boldsymbol{Y}_i; \mathcal{W}]_i\big) \tag{51}$$

obviously follows.

## C   EXPERIMENTAL DETAILS

### C.1   TRAINABLE LABEL PROPAGATION

In Table 1, we employ trainable label propagation with the same settings as the original label propagation algorithm, including the trade-off term $\lambda$ in (1) and the number of propagation steps. In the implementation, we set $\lambda$ as 0.6 and use 50 propagation steps. We search the best splitting probability $\alpha$ in the range of $\{0.05, 0.1, \ldots, 0.95\}$.

### C.2   DETERMINISTIC LABEL TRICK APPLIED TO GNNS WITH LINEAR LAYERS

We use three models with linear propagation layers and simply choose one specific set of hyperparameters and run the model on each dataset with or without label trick in Table 3. The hyperparameters for each model is described as follows. For TWIRLS, we use 2 propagation steps for the linear propagation layers and $\lambda$ is set to 1 (Yang et al., 2021). For SIGN, we sum up all immediate results instead of concatenating them to simplify the implementation and speed up training. Our SIGN model also use 5 propagation steps, and we tune the number of MLP layers from 1 or 2 on each dataset. And for SGC, the number of propagation steps is set to 3, and there is one extra linear transformation after the propagation steps.

### C.3   EFFECT OF SPLITTING PROBABILITY

To further present the effect of splitting probability $\alpha$, in Figure 1, we use $\alpha$ in the range of $\{0.00625, 0.0125, 0.025, 0.05, 0.1, 0.2, 0.3, 0.4, 0.5, 0.6, 0.7, 0.8, 0.9, 0.95, 0.975, 0.9875, 0.99375\}$ to compare a linear model (similar to a trainable label propagation algorithm with features and labels as input) with a three-layer GCN (Kipf & Welling, 2017). Each model uses the same set of hyperparameters, except for the number of epochs, since when $\alpha$ is close to zero or one, the model requires more epochs to converge. For the linear model, $\lambda$ is set to 0.9 and the number of propagation steps is 9. The GCN has 256 hidden channels and an activation function of ReLU.

## C.4 TRAINABLE CORRECT AND SMOOTH

We begin by restating the predictor formulation of trainable Correct and Smooth in (14):

$$
\begin{aligned}
f_{TC\&S}(\tilde{\boldsymbol{Y}}; \mathcal{W}) &= \boldsymbol{P}_s(\boldsymbol{Y}_{in} + (\boldsymbol{M}_{te} + \boldsymbol{M}_{out})(\tilde{\boldsymbol{Y}} + \tilde{\boldsymbol{E}}_{in}\boldsymbol{W}_c))\boldsymbol{W}_s \\
&= \boldsymbol{P}_s((\boldsymbol{Y}_{in} + (\boldsymbol{M}_{te} + \boldsymbol{M}_{out})\tilde{\boldsymbol{Y}})\boldsymbol{W}_s + \boldsymbol{P}_s(\boldsymbol{M}_{te} + \boldsymbol{M}_{out})\tilde{\boldsymbol{E}}_{in}\boldsymbol{W}_c\boldsymbol{W}_s \\
&= \underbrace{\boldsymbol{P}_s((\boldsymbol{Y}_{in} + (\boldsymbol{M}_{te} + \boldsymbol{M}_{out})\tilde{\boldsymbol{Y}})}_{\hat{\boldsymbol{Y}}_s}\boldsymbol{W}_s + \underbrace{\boldsymbol{P}_s(\boldsymbol{M}_{te} + \boldsymbol{M}_{out})\tilde{\boldsymbol{E}}_{in}}_{\hat{\boldsymbol{Y}}_c}\hat{\boldsymbol{W}}_c,
\end{aligned}
\tag{52}
$$

Note that the computation of $\tilde{\boldsymbol{Y}}_s$ and $\tilde{\boldsymbol{Y}}_c$ does not involve any trainable parameters. Therefore, we can compute them beforehand for each training split.

We now describe our experimental setup used to produce Table 4. We first trained an MLP with exactly the same hyperparameters as in Huang et al. (2021). For each splitting probability $\alpha \in \{0.1, 0.2, \cdots, 0.9\}$, we generate 10 splits and precompute $\tilde{\boldsymbol{Y}}_s$ and $\tilde{\boldsymbol{Y}}_c$. Then for each training epoch, we cycle over the set of $\tilde{\boldsymbol{Y}}_s$ and $\hat{\boldsymbol{Y}}_c$.

