# OpenReview forum: "Why Propagate Alone? Parallel Use of Labels and Features on Graphs"
_ICLR.cc/2022/Conference — ICLR 2022 Poster_

### Official Review · Reviewer_n4P3 · 2021-10-25

**Correctness:** 4
**Technical Novelty And Significance:** 3
**Empirical Novelty And Significance:** 2
**Recommendation:** 5
**Confidence:** 4

**Main Review:**

Strength:
- It is nice to take a step back to tries to understand what might look like wizard engineering, and show that the labels tricks can be understood with some simple models.
- This paper is relatively easy to follow and well-written

Weaknesses:
- This results in this paper are not really satisfactory. The background and motivations are not really well explained. And the duality between stochasticity of the label trick and the deterministic regularization is not really back-up by "to-the-point" experiments.
- The paper is lacking rigor as I will detail later.
- Some claims of the paper are weird to me, in particular, the way the graph is built should indeed use features knowledge. So graph-based Laplacian do leverage features (yet in an unsupervised way). Also if the node features are noisy or unreliable, which will downgrade the performance of GNN, I believe the graph structure than will also be noisy or unreliable for those nodes if the graph is build based on features similarity.
- For a paper that seeks a theoretical explanation of the usefulness of the label trick. I do not have the impression to have learn a lot from it. For example, why self-excluded prediction do not work as well as full execution of the label trick?
Also, why the label trick is better than GNN with a Laplacian regularization akin Pfau et al. or Cabannes et al. with a MixUp regularization akin Lopez-Paz et al.?


Notes

page 1:
1/ The readibility of the paper would be improved if the authors were to go straight to the point and more rigorous. For example of “straight to the point", at line 4 in the introduction, the fact that labels and features vary smoothly over the graph and i.i.d. are two differents problems. And talking about i.i.d. samples is only adding confusion in the mind of the reader (at least in my mind).
For example of “rigorous", in the abstract, talking about the identity mapping from input to output, I believe the authors talks about the transductive setting, in which the feature X is forgotten but a propagation rule is used, which can be null as explained for label leakage in Eq. (3). Clearly the "identity mapping" from inputs in R^d to outputs in R^c is not well defined.

page 2:
1/ It would be nice to explain more clearly the graph formalism. For example:
Each node is linked to a input point x in R^d, and a label in R^c, from which we can form the node feature matrix (or design matrix) X and the label matrix of the nodes (or label matrix) Y. The graph is a undirected and edge have weights. The adjacency matrix is the matrix of weights (symmetric because the graph is undirected), thus S is symmetric.

2/ "Nodes connected by an edge […] are likely to share the same labels". If the graph is weighted this depend on the weight. If the graph is unweighted, it should be specified.

3/ Eq.(1) I believe the second F should be F_tr, otherwise, there is, using pythagoras a penalization pushing F towards zeros on testing data. A close form solution can still be written with this correct formulation as well as the power method. The fact that Y = 0 is an informative labels (e.g. because Y are logits) and that pushing F towards 0 does not bias the system should be specified.
Also it would be nice to specify F \in R^{n \times c}, and to specify also that this minimizing this objective can be done independently according to each coordinates of \Y = \R^c, if there is no constraints on F.

page 3:
1/ There is a mistake with P \approx PY on the second paragraph.
Also the approximation can be written as an equality with an inifinite sum since \lambda S is contracting (equality with a series being here understood as the convergence in any norm).

2/ "where […] P smooths the training labels” is weird since P is not used to defined F^(k). This observation should be written after the definition of P = (1 - \lambda)(I - \lambda S)^{-1}.

page 4:
1/ "model from (…)”: the parenthesis should be removed

2/ A note on SGC would be highly appreciated, as well as an explanation on the relevance of such a model. Space can be found by condensing section 3.2. and 3.3.

3/ What is P in Eq.(3), P was defined as “a propagation matrix”, how is it fixed here?

Appendix:
There is several overboxes, please check the log file when compiling latex.

References formatting:
ArXiv -> journal and conference publications


**Summary Of The Paper:**

The paper focus on the label trick.
The label tricks is a method for graph-based semi-supervised learning.
In this setting, we have some labeled data points and some unlabeled data points, the goal is to label the unlabeled data.
The data points can form vertices of a graph, with edges built based on similarity of the points (eventually with a weighting scheme).
Labels can be diffused along the graph to label unlabeled data thanks to graph Laplacian.
A other way to leverage a graph structure is to build a graph neural networks and learn parameters of this networks to match labels of labeled data.
The authors states that the first method is utilizing labels and the second method is utilizing features based on a graphical model. They say that both should be done simultaneously, and argue that this is what the label trick is doing.

Eventually known labels can used along with features as the input of a GNN, but if we only ask to match labels of labeled data, than a trivial mapping can be learnt (the identity from known labels to the labels of labeled data). Therefore it is smart to use a drop-out strategy where we hide few labels (in particular the ground-truth label) when we want to learn the labels of a labeled data. This is the label trick.

The paper tries to understand what the label trick is about by looking at simple models.
It also discussed on its novel-case of applications.


**Summary Of The Review:**

Overall while having justifications of what people do in practice is nice, I think this paper fall short of providing a satisfactory analysis of the label tricks, especially since it does not rigorously explained it superiority compared to other methods.
I think this stream of research is valuable but this is too early for publication.

---

> ### Author Response · Authors · 2021-11-18
> **Response to Reviewer n4P3 Part 1**
>
> Thanks for the detailed comments. We have also now posted a new version of the paper which resolves some of the issues that were raised regarding the original submission, and includes one new technical result (Theorem 3).
>
> **Question:** This results in this paper are not really satisfactory.
>
> **Response:** The reviewer has questioned the significance of the results in Tables 1-4.  We address these results table by table.
>
> *Table 1*: The label trick already demonstrates statistically significant results on both Arxiv and Products in Table 1. Moreover, the Cora-full results are also significant once we account for the fact that the higher variance mainly comes from the random division of the dataset into training, validation and test dataset.  In this way, the performance gap remains consistent across trials such that the improvement is actually still significant. For example, the following table shows that the label trick consistently improves the results in Table 1 on Cora-full over different dataset splits even after careful tuning of hyperparameters on the baseline model.  Hence it is only on Pubmed where only a small, possibly negligible advantage is displayed.
>
> Table: Accuracy results on Cora-full.
>
> |  Trial  | Label Propagation | Trainable Label Propagation |
> |:-------:|:-----------------:|:---------------------------:|
> |    1    |       65.52       |          **67.57**          |
> |    2    |       65.77       |          **66.03**          |
> |    3    |       64.84       |          **66.28**          |
> |    4    |       66.00       |          **66.23**          |
> |    5    |       68.10       |          **68.58**          |
> |    6    |       67.39       |          **67.87**          |
> |    7    |       67.29       |          **68.07**          |
> |    8    |       66.86       |          **68.83**          |
> |    9    |       66.48       |          **67.85**          |
> |   10    |       66.13       |          **66.73**          |
> | Average |   66.44 ± 0.93    |      **67.40 ± 0.96**       |
>
> *Table 2*: All improvements except for the Avazu dataset are significant once we account for shared, trial-to-trial variability (in the same fashion as we showed for Cora-full above).
>
> *Table 3*:  These results are meant to highlight broader use cases of the deterministic simplification of the label trick.  And the Arxiv results do show significant improvement in Table 3; however, for both Cora-full and Pubmed the performances with and without the label trick are indeed similar.  As mentioned in the paper (end of Section 5.2), this is because for these datasets under the current experimental settings, the training accuracy is near 100% such that the label information is already adequately embedded in the model and the label trick has limited space for impact.
>
> To further illustrate the role of training accuracy in training with the label trick, we produce two new datasets by projecting the input features on Cora-full and Pubmed to 32 dimensions via PCA. This is a much more challenging task because less feature information is available for prediction. We train SGC, SIGN and TWIRLS under the same settings as in Table 3. The following tables show the test and training accuracy on these datasets respectively, as well as results using original features as in Table 3.
>
> Table: Test accuracy with/without label trick (D).
>
> |       Dataset       |       SGC        |       SGC        |       SIGN       |       SIGN       |    TWIRLS    |      TWIRLS      |
> |:-------------------:|:----------------:|:----------------:|:----------------:|:----------------:|:------------:|:----------------:|
> | **label trick (D)** |        ✗         |        ✓         |        ✗         |        ✓         |      ✗       |        ✓         |
> |    Cora-full-32d    |   55.63 ± 0.53   | **62.37 ± 0.87** |   55.10 ± 0.55   | **62.05 ± 0.47** | 58.57 ± 0.48 | **61.74 ± 0.46** |
> |     Pubmed-32d      |   83.15 ± 0.41   | **84.33 ± 0.68** |   84.75 ± 0.37   | **85.72 ± 0.84** | 84.94 ± 0.37 | **87.60 ± 0.44** |
> |      Cora-full      | **65.87 ± 0.61** |   65.81 ± 0.69   | **68.54 ± 0.76** |   68.44 ± 0.88   | 70.36 ± 0.51 | **70.40 ± 0.71** |
> |       Pubmed        |   85.02 ± 0.43   | **85.23 ± 0.57** |   87.94 ± 0.52   | **88.09 ± 0.59** | 89.81 ± 0.56 | **90.08 ± 0.52** |

---

> > ### Comment · Reviewer_n4P3 · 2021-11-21
> > **Questions unanswered**
> >
> > I thank the authors for their response.
> >
> > Some of my remarks were not properly answered, so I benefit of this phase of discussion to mention them.
> >
> > The fact that Y = 0 is an informative labels (e.g. because Y are logits) and that pushing F towards 0 does not bias the system should be specified.
> >
> > The formatting of the references was not changes from ArXiv to NeurIPS for "Open Graph Benchmark: Datasets for Machine Learning on Graphs” contrarily to what the authors wrote in the rebuttal!!!
> >
> > For the references, I was thinking that you were working on problem like semi-supervised learning, where the graph is built as a model that help to learn a function from input to output, as such I was mentioning other modeling technics that are not based on graphs, but still on label propagation based on features structure.
> > As I said, the motivations were not clear to me in the first draft. Now I see that you put yourself at a stage where the graph is already given.
> > Agree that sentences like "Label propagation is a semi-supervised algorithm that predicts unlabeled nodes by propagating the observed labels across the edges of the graph” are confusing as you are rather motivated by a transductive setting than a semi-supervised learning setting (where the goal is to learn a generic mapping from features to labels).
> >
> > I really do not agree with you on this non iid thing. In machine learning we suppose that samples (X, Y) are drawn in an iid manner, but we also suppose that (Y | X) is a function of X, such that if you build a graph from samples as in the seminal work of Zhu et al on graph Laplacian, then you have dependency between labels in the graph, motivating the use of graph Laplacian. However, in this work there is no mention that nodes labels are conditionally independent regarding nodes that connect them (as in graphical model), so I do not see the fact that if there is no edges then everything is iid, which you mentioned in the rebuttal.
> >
> > I do not want to seem harsh with the authors, and I value the work that there is behind this paper. My comments are meant to be constructive, and I really talk from a reader perspective who wants to get the most out of this great work, and wants to benefit from an advanced understanding of technics that work in practice. But as it is, I believe this work need more clarifications.

---

> > > ### Author Response · Authors · 2021-11-22
> > > **Response to Reviewer n4P3 Part 1**
> > >
> > > Thanks for the reply. We will address each one below.
> > >
> > > **Question:** The fact that Y = 0 is an informative labels (e.g. because Y are logits) and that pushing F towards 0 does not bias the system should be specified.
> > >
> > > **Response:** Perhaps we misunderstood the reviewer's question here, but our work does not attempt to motivate or justify the design choices of the basic label propagation algorithm.  Instead, for background context we merely present the established, most common version that exists in the literature, and then later demonstrate how the label trick can be applied to improving results.  Specific biases introduced by the assumption Y = 0 for unlabeled nodes is beyond our scope and unrelated to our analysis of the label trick.  Incidentally, the label trick can also be equally applied to other less-common LP variants if so desired, but we believe our initial proof-of-concept with the standard form is adequate for present purposes.
> > >
> > > **Question:** The formatting of the references was not changes from ArXiv to NeurIPS for "Open Graph Benchmark: Datasets for Machine Learning on Graphs” contrarily to what the authors wrote in the rebuttal!!!
> > >
> > > **Response:** We previously reformatted all the references that we were aware needed updating.  However, in terms of the specific reference the reviewer mentioned, namely, "Open Graph Benchmark: Datasets for Machine Learning on Graphs," we originally obtained the citation from the OGB website itself https://ogb.stanford.edu/, which directly links to the arXiv version, not the NeurIPS paper.  We apologize that we did not catch this earlier, but have done so now and uploaded a revision to OpenReview.  (As a minor side note though, we actually suspect that the authors of the OGB paper may possibly prefer citation of the arXiv version since there have been multiple updates since the NeurIPS conference, so now the official NeurIPS version is presumably outdated.)
> > >
> > > **Question:** For the references, I was thinking that you were working on problem like semi-supervised learning, where the graph is built as a model that help to learn a function from input to output, as such I was mentioning other modeling technics that are not based on graphs, but still on label propagation based on features structure. As I said, the motivations were not clear to me in the first draft. Now I see that you put yourself at a stage where the graph is already given. Agree that sentences like "Label propagation is a semi-supervised algorithm that predicts unlabeled nodes by propagating the observed labels across the edges of the graph” are confusing as you are rather motivated by a transductive setting than a semi-supervised learning setting (where the goal is to learn a generic mapping from features to labels).
> > >
> > > **Response:**  Thanks for the clarification.  One quick follow-up point though: semi-supervised learning includes both transductive and inductive learning.  Indeed, even one of the seminal papers on trainable deep graph models for tranductive tasks is called "Semi-supervised classification with graph convolutional networks" by Kipf and Welling.  That being said, we can double-check the updated draft to ensure there is minimal ambiguity with respect to this topic.
> > >
> > > **Question:** I really do not agree with you on this non iid thing. In machine learning we suppose that samples (X, Y) are drawn in an iid manner, but we also suppose that (Y | X) is a function of X, such that if you build a graph from samples as in the seminal work of Zhu et al on graph Laplacian, then you have dependency between labels in the graph, motivating the use of graph Laplacian. However, in this work there is no mention that nodes labels are conditionally independent regarding nodes that connect them (as in graphical model), so I do not see the fact that if there is no edges then everything is iid, which you mentioned in the rebuttal.
> > >
> > > **Response:**  This distinction between iid vs non-iid data only appears in one sentence in our paper as background context regarding the influence of a pre-specified graph structure.  While similar descriptions are common in the GNN literature, we are happy to remove or change if the intended meaning is unclear.  With respect to the latter, the point we were trying is perhaps even much simpler than the situation the reviewer described.  By non-iid we simply meant that the order of the samples in the dataset matter as they must be aligned with a *pre-defined* graph (not a graph derived from the samples themselves).  In other words, we cannot arbitrarily shuffle the node labels/features relative to this fixed graph, as would be the case for standard iid DNN training of something like imagenet where there is no such graph (and the sample order is inconsequential).
> > >
> > > Regardless, if this usage seems misleading or unclear to the reviewer, we can easily revise the draft further to clarify.  This would be a simple fix.

---

> > > > ### Comment · Reviewer_n4P3 · 2021-11-22
> > > > **On references**
> > > >
> > > > I appreciate the authors response.
> > > >
> > > > But regarding references, I took a random one in my previous comment. The authors say that unfortunately this is the only one they miss.
> > > > Now I take another random one "Predict then Propagate: Graph Neural Networks meet Personalized PageRank", and once again I see that this paper was presented at ICLR 2019, but the authors cite the ArXiv version.
> > > > I do not mind if the authors want to argue that referencing arxiv is easier than referencing conferences and journals, and that they do not want to spend too much time reformatting references, but I do mind if they told me that they have changed the format, while they have not done so. I am really confused by this behavior.

---

> > > > > ### Author Response · Authors · 2021-11-22
> > > > > **Response to Reviewer n4P3**
> > > > >
> > > > > We apologize for the oversight here.  This is a collaborative project among multiple authors and there was a miscommunication within our team regarding whether or not all of the references had been checked.  Note that we also just noticed this mistake on our side and have already uploaded the revision.

---

> > > > > > ### Comment · Reviewer_n4P3 · 2021-11-22
> > > > > > **On reference format**
> > > > > >
> > > > > > Great, in term of formatting, it is better to stick to a still convention, here you have multiple format (full conference name + location; or full conference name without locations; or small conference name).
> > > > > > But this is details that most people do not really care about.

---

> > > > > > > ### Author Response · Authors · 2021-11-22
> > > > > > > **Response to Reviewer n4P3**
> > > > > > >
> > > > > > > Good suggestion.  We have removed some of the less relevant publication details and used more consistent formatting.

---

> > > > > > > ### Author Response · Authors · 2021-11-26
> > > > > > > **To Reviewer n4P3**
> > > > > > >
> > > > > > > We hope our comments have adequately addressed the reviewer's previous concerns. Given the limited remaining time of the rebuttal session, we wonder if the reviewer has any more feedback?

---

> > > ### Author Response · Authors · 2021-11-22
> > > **Response to Reviewer n4P3 Part 2**
> > >
> > > **Question:** I do not want to seem harsh with the authors, and I value the work that there is behind this paper. My comments are meant to be constructive, and I really talk from a reader perspective who wants to get the most out of this great work, and wants to benefit from an advanced understanding of technics that work in practice. But as it is, I believe this work need more clarifications.
> > >
> > > **Response:** We genuinely appreciate the reviewer's effort to provide constructive feedback and for wanting to get the most out of our work.  These discussions can certainly improve the readability and help make the content accessible to a wider audience.  And hopefully the updated draft and responses above help to clarify any of the lingering uncertainty.

---

> ### Author Response · Authors · 2021-11-18
> **Response to Reviewer n4P3 Part 2**
>
> Table: Training accuracy with/without label trick (D).
>
> |       Dataset       |     SGC      |       SGC        |     SIGN     |       SIGN       |      TWIRLS      |      TWIRLS      |
> |:-------------------:|:------------:|:----------------:|:------------:|:----------------:|:----------------:|:----------------:|
> | **label trick (D)** |      ✗       |        ✓         |      ✗       |        ✓         |        ✗         |        ✓         |
> |    Cora-full-32d    | 57.69 ± 0.21 | **65.75 ± 0.19** | 57.20 ± 0.21 | **65.88 ± 0.27** |   63.44 ± 0.41   | **67.40 ± 0.36** |
> |     Pubmed-32d      | 83.43 ± 0.35 | **84.74 ± 0.17** | 84.96 ± 0.13 | **86.05 ± 0.19** |   85.97 ± 0.22   | **88.43 ± 0.26** |
> |      Cora-full      | 95.31 ± 0.16 | **97.59 ± 0.12** | 99.78 ± 0.03 | **99.99 ± 0.01** | **99.95 ± 0.01** |   99.76 ± 0.05   |
> |       Pubmed        | 85.63 ± 0.17 | **85.69 ± 0.23** | 91.26 ± 0.12 | **91.43 ± 0.17** |   97.32 ± 0.30   | **97.48 ± 0.28** |
>
> From the two tables above, we can see that the degree to which the label trick can improve accuracy is directly related to the training accuracy. When the training accuracy is near 100%, the label trick could hardly benefit the model as the label information is already adequately embedded in the model. However, when the training accuracy is lower, the label trick can increase both the training and test accuracy significantly (once the variance from the dataset divisions is taken into account).
>
> Moreover, for large-scale graphs it may be less likely that high accuracy or overfitting is obtained during training due to the large number of training samples.  So in such cases, the label trick is more likely to be effective.
>
> *Table 4*: Although a few results are comparable to those without the label trick, our main goal is to present a novel use case and to show that C&S is parameterizable. However, we have only used the simplest linear model here, which has been able to achieve decent results. This motivates its application to more complex cases, including nonlinear models, where regular C&S cannot perform.
>
> Regardless, the most important contribution of our paper is not promoting a new technique per se, but rather, to provide a theoretical foundation for the label trick, while demonstrating some previously-untried applications of the label trick that can motivate further use cases.
>
> **Question:** The background and motivations are not really well explained. And the duality between stochasticity of the label trick and the deterministic regularization is not really back-up by "to-the-point" experiments.
>
> **Response:** The revised version uploaded to OpenReview provides additional clarification; however, we will address each specific issue the reviewer raised point-by-point below.  Additionally, Figure 1 (left) provides a comparison between the stochastic label trick and the self-excluded, deterministic simplification.
>
> **Question**: The paper is lacking rigor as I will detail later.
>
> **Response:** We emphasize that, while the reviewer requested additional clarity in certain places (see below), the stated lack of rigor did not include any challenge to our main theoretical results.
>
> **Question**: Some claims of the paper are weird to me, in particular, the way the graph is built should indeed use features knowledge. So graph-based Laplacian do leverage features (yet in an unsupervised way). Also if the node features are noisy or unreliable, which will downgrade the performance of GNN, I believe the graph structure than will also be noisy or unreliable for those nodes if the graph is build based on features similarity.
>
> **Response:** The basic assumption of most GNN benchmark datasets is that the graph structure is given/fixed, and may or may not include noisy features.  And importantly, the graph structure itself need not be built based on feature similarity.  In fact, in many situations the opposite may be the case, whereby features are learned or heuristically constructed for a given graph.  Regardless, these issues are mostly orthogonal to our central contributions, which pertain to obtaining improved predictions by combining the strengths of both label and feature propagation.  How the actual graphs were originally constructed is beyond the scope of our work.

---

> ### Author Response · Authors · 2021-11-18
> **Response to Reviewer n4P3 Part 3**
>
> **Question**: For a paper that seeks a theoretical explanation of the usefulness of the label trick. I do not have the impression to have learn a lot from it. For example, why self-excluded prediction do not work as well as full execution of the label trick? Also, why the label trick is better than GNN with a Laplacian regularization akin Pfau et al. or Cabannes et al. with a MixUp regularization akin Lopez-Paz et al.?
>
> **Response:**  As detailed in Section 3, full execution of the stochastic label trick amounts to a regularized version of the self-excluded trick, and this adaptive, graph-dependent regularization can provide benefits.  Moreover, the label trick can be applied to most any GNN model, with or without additional regularization factors.  Note also that standard Laplacian regularization in a typical GNN context does not lead to the self-excluded model, nor any dependency on labels.
>
> And finally, we apologize, but it was somewhat unclear to us which specific references the reviewer is referring to.  After searching a bit, possible candidates include "Overcoming the curse of dimensionality with Laplacian regularization in semi-supervised learning" by Cabannes et al., and "Mixup: Beyond Empirical Risk Minimization" by Lopez-Paz et al.  However, these references do not discuss issues closely-related to the label trick or our attendant analysis.  Could the reviewer point us to specific references?  If so, we would be happy to review and provide relevant discussion/comparison.
>
>
> **Question**: page 1: 1/ The readibility of the paper would be improved if the authors were to go straight to the point and more rigorous. For example of “straight to the point", at line 4 in the introduction, the fact that labels and features vary smoothly over the graph and i.i.d. are two differents problems. And talking about i.i.d. samples is only adding confusion in the mind of the reader (at least in my mind). For example of “rigorous", in the abstract, talking about the identity mapping from input to output, I believe the authors talks about the transductive setting, in which the feature X is forgotten but a propagation rule is used, which can be null as explained for label leakage in Eq. (3). Clearly the "identity mapping" from inputs in R^d to outputs in R^c is not well defined.
>
> **Response:** If we remove all edges from the graph, then the problem reduces to a typical i.i.d. ML setting; however, the graph naturally introduces non-i.i.d. dependencies between samples that would otherwise not exist.  This is the primary motivation for GNN models. Regardless, the abstract has been completely re-written and we believe it is now much more clear. Furthermore, to further clarify here, the transductive setting does not mean that features X are ignored; rather, it simply means that during training we have access to all of the test nodes including their node features and edges.
>
>
> **Question**: page 2: 1/ It would be nice to explain more clearly the graph formalism. For example: Each node is linked to a input point x in R^d, and a label in R^c, from which we can form the node feature matrix (or design matrix) X and the label matrix of the nodes (or label matrix) Y. The graph is a undirected and edge have weights. The adjacency matrix is the matrix of weights (symmetric because the graph is undirected), thus S is symmetric.
>
> **Response:** We have very limited space to provide further background details regarding graphs; however, we have added a few more details in the revision, and for a final version we can try squeezing in further descriptions where possible.
>
>
> **Question:** 2/ "Nodes connected by an edge […] are likely to share the same labels". If the graph is weighted this depend on the weight. If the graph is unweighted, it should be specified.
>
> **Response:**  The graph is unweighted, and we have specified this in the revision.

---

> ### Author Response · Authors · 2021-11-18
> **Response to Reviewer n4P3 Part 4**
>
> **Question:** 3/ Eq.(1) I believe the second F should be F_tr, otherwise, there is, using pythagoras, a penalization pushing F towards zeros on testing data. A close form solution can still be written with this correct formulation as well as the power method. The fact that Y = 0 is an informative labels (e.g. because Y are logits) and that pushing F towards 0 does not bias the system should be specified. Also it would be nice to specify F \in R^{n \times c}, and to specify also that this minimizing this objective can be done independently according to each coordinates of \Y = \R^c, if there is no constraints on F.
>
> **Response:** Actually $F$ (not $F_{tr}$) is correct. This is the widely-adopted label propagation setting with the same energy function as in Zhou et al., "Learning with local and global consistency," Yang et al., "Graph Neural Networks Inspired by Classical Iterative Algorithms," Jia et al., "A unifying generative model for graph learning algorithms: Label propagation, graph convolutions, and combinations", which indeed can push some rows of $\mathbf F$ towards near $0$. That being said, alternative label propagation use cases could be motivated using $\mathbf F_{tr}$ not $\mathbf F$. However, this is beyond our scope.
>
> **Question:** page 3: 1/ There is a mistake with P \approx PY on the second paragraph. Also the approximation can be written as an equality with an inifinite sum since \lambda S is contracting (equality with a series being here understood as the convergence in any norm). 2/ "where […] P smooths the training labels” is weird since P is not used to defined F^(k). This observation should be written after the definition of P = (1 - \lambda)(I - \lambda S)^{-1}.
>
> **Response:** We have made this paragraph clearer in the revision.
>
> **Question:** page 4: 1/ "model from (…)”: the parenthesis should be removed
>
> **Response:** We have fixed all formatting issues of citation, please see our revised version.
>
> **Question:** 2/ A note on SGC would be highly appreciated, as well as an explanation on the relevance of such a model. Space can be found by condensing section 3.2. and 3.3.
>
> **Response:** SGC is one of the mostly popular GNN models; however, for present purposes it is sufficient to know that SGC adopts linear predictors of the form $\mathbf P \mathbf X \mathbf W$, where $\mathbf P$ is a graph propagation operator, $\mathbf X$ are the node features, and $\mathbf W$ are model weights.
>
> **Question:** 3/ What is P in Eq.(3), P was defined as “a propagation matrix”, how is it fixed here?
>
> **Response:** $\mathbf P$ in equation (3) can be any arbitrary propagation matrix, e.g., the normalized graph Laplacian or the $\mathbf P$ described in Section 2.1, etc.
>
> **Question:** Appendix: There is several overboxes, please check the log file when compiling latex. References formatting: ArXiv -> journal and conference publications
>
> **Response:** We have fixed these formatting issues, please see the revision.
>
> **Question:** Overall while having justifications of what people do in practice is nice, I think this paper fall short of providing a satisfactory analysis of the label tricks, especially since it does not rigorously explained it superiority compared to other methods. I think this stream of research is valuable but this is too early for publication.
>
> **Response:** Our work does actually help to rigorously explain how the label trick can allow us to successfully propagate both features and labels, and given its widespread practical adoption, we believe this to be an important contribution.  Moreover, this analysis directly motivates new variants of the label trick.

---

### Official Review · Reviewer_bkGb · 2021-10-31

**Correctness:** 3
**Technical Novelty And Significance:** 3
**Empirical Novelty And Significance:** 3
**Recommendation:** 5
**Confidence:** 4

**Main Review:**

### Strengths (+) & Weaknesses (–)

(+) The problem that the authors tackle is fundamental/important. The community would benefit from research that connects the ideas of classical label propagation methods and message passing neural networks.

(+)  The approach is well-motivated in that one main issue of the label trick is the undesirable solution of the identity mapping, but only keeping off-diagonal terms in the propagation matrix is a seemingly simple fix (whereas diagonal terms are fine for feature propagation). That is to say, Equation (10) makes intuitive sense.

(+) The work expounds upon the use of the label trick, and is right in saying that it has been mainly used as a heuristic, seeing effective use. Studying it further is a worthwhile pursuit.

(–) Some further emphasis/credit should be given to https://arxiv.org/pdf/2009.03509.pdf who (I believe) first proposed the label trick.

(–) What is the intuition behind equation (3), namely why would one want to express the column space of $Y_{tr}$ by post-multiplying with a weight matrix $W$? I wonder similarly for the trainable Smoothing step in Correct & Smooth.

(–) Given that the paper has some 'theoretical flavor' to it, we should see some comparison to other works that compare label and feature propagation (even if these propagations are not used in parallel), e.g., Jia & Benson 2021 (https://arxiv.org/pdf/2101.07730.pdf).

(–) Some equations are unclear, and the subscript of $\mathbb{E}_{splits}$ is vague/difficult to parse. I would appreciate some elaboration on Equation 5 in particular, how does one go from the second-last term to the last term? Since D_out has cardinality 1, should the expectation not be over a discrete uniform sample from $i \sim \{ 1 \dots n \}$, which would simplify to

$ \frac{1}{n} \sum_{i \in D_{tr}} \ell (y_i, [P(Y_{tr} - Y_i W]_i ) $.

Could you please tell me what I'm missing/failing to see here? Thanks.

More generally, the paper is at times hard to read due to it being notationally dense (particularly the excessive use of subscripts), and the stream of the 'Remarks'. I feel the paper could benefit from one more pass of re-writing.

(–) For Table 2, R^2 value should be used to evaluate regression.

(–) Section 4.1 is unclear and should have greater detail. For example, how is this procedure trained? What is the final model (is it just supervision on the Neumann series with a weight matrix instead of a propagation matrix)? Further, some reference should be given to other papers who learn/modify the adjacency/propagation matrix.

### Questions:

- Why isn't the deterministic label trick used for comparison in the right plot in Figure 1? Further, which GNN is used? I believe this is unspecified in Section 5.3.
- In remark 4, if the regularization is shut off why is there "no chance for overfitting"?
- If the derivation shows that we can eventually decompose with a regularization term in the deterministic case, is this used for results in Table 3? If it is not used (as it does not seem to be mentioned), why not?

Some less important nitpicks:
- typo in remark 2 "insure" → "ensure"
- inconsistency in use of parentheses vs square brackets in (4) and (5)


**Summary Of The Paper:**

The authors discuss the utilization of labels in graph neural networks (or conversely, the utilization of node features in collective classification algorithms, e.g., label propagation) for node prediction. It expounds on the stochastic 'label trick' whereby training labels are concatenated to node features and are then randomly masked to avoid trivial solutions. With simplifying assumptions (where there are no node features), they show that this procedure is reduced to minimizing fitting and regularization terms, which is used as motivation for a few practical methods.


**Summary Of The Review:**

The paper has strengths and tackles an important problem, but needs further work on the points mentioned above.

---

> ### Author Response · Authors · 2021-11-18
> **Response to Reviewer bkGb Part 1**
>
> Thanks for the detailed comments. We have now posted a new version of the paper which resolves some of the issues that were raised regarding the original submission, and includes one new technical result (Theorem 3).  Below we also respond point-by-point to each reviewer comment.
>
> **Question:** Some further emphasis/credit should be given to https://arxiv.org/pdf/2009.03509.pdf who (I believe) first proposed the label trick.
>
> **Response:** Our submission does not claim to originate the label trick and repeatedly refers to its widespread prior use in the literature.  That being said, the first instance in the literature does appear to be the arxiv paper the reviewer mentioned.  However, we first encountered the label trick from a github post on the OGB leaderboard that predates the above paper and has been adopted by many other groups/leaderboard submissions.  Hence we have merely cited all the available prior literature on the label trick that we are aware of without explicitly specifying an original source.
>
> **Question:** What is the intuition behind equation (3), namely why would one want to express the column space of $Y_{tr}$ by post-multiplying with a weight matrix $W$? I wonder similarly for the trainable Smoothing step in Correct & Smooth.
>
> **Response:** Equation (3) can be viewed and motivated as a simplification of Definition 1, whereby there are no node features and the function $f$ is constrained to be a single linear layer model.  Additionally, the weight matrix $\mathbf W$ in (3) can be further justified in situations where the class labels are interrelated (i.e., not independent) as discussed in Section 4.1.
>
> **Question:** Given that the paper has some 'theoretical flavor' to it, we should see some comparison to other works that compare label and feature propagation (even if these propagations are not used in parallel), e.g., Jia & Benson 2021 (https://arxiv.org/pdf/2101.07730.pdf).
>
> **Response:** The recent arxiv paper the reviewer mentions is indeed relevant, although it does not focus on the parallel propagation of labels and features or the analysis of the label trick as we do. Still, it's a great reference that we have added to the revision.
>
> **Question:** Some equations are unclear, and the subscript of $\mathbb E_{splits}$ is vague/difficult to parse. I would appreciate some elaboration on Equation 5 in particular, how does one go from the second-last term to the last term? Since $D_{out}$ has cardinality 1, should the expectation not be over a discrete uniform sample from $i \sim 1\ldots n$, which would simplify to
> $$\frac{1}{n}\sum_{i\in D_{tr}}\ell(y_i,[P(Y_{tr}-Y_{i})W]_i).$$
>
> Could you please tell me what I'm missing/failing to see here? Thanks.
>
> **Response:** $\mathbb E_{splits}$ denotes the expectation over the splits of the training set (i.e., $\mathcal D_{in}$ and $\mathcal D_{out}$) as described in Definition 1. Regarding the second question, here is the detailed derivation of this equation. Let $\mathbf M_i$ be the diagonal matrix whose i-th row, i-th column, is $1$ and all others are $0$. Note that $[\mathbf A]_i=[\mathbf M_i\mathbf A]_i$ holds for any matrix $\mathbf A$, and $\mathbf M_i\mathbf P\mathbf M_i=\mathbf M_i\mathbf C\mathbf M_i=\mathbf M_i\mathbf C$, therefore,
>
> $$
> \begin{aligned}
> &\frac{1}{n}\sum\_{i\in D\_{tr}}\ell(\mathbf y\_i,[\mathbf P(\mathbf Y\_{tr}-\mathbf Y\_{i})\mathbf W]\_i)\\\\
> &=\frac{1}{n}\sum\_{i\in D\_{tr}}\ell(\mathbf y\_i,[\mathbf P\mathbf Y\_{tr}\mathbf W]\_i-[\mathbf P\mathbf Y\_{i}\mathbf W]\_i)\\\\
> &=
> \frac{1}{n}\sum\_{i\in D\_{tr}}\ell(\mathbf y\_i,[\mathbf P\mathbf Y\_{tr}\mathbf W]\_i-[\mathbf M\_i\mathbf P\mathbf M\_i\mathbf Y\_{tr}\mathbf W]\_i)\\\\
> &=
> \frac{1}{n}\sum\_{i\in D\_{tr}}\ell(\mathbf y\_i,[\mathbf P\mathbf Y\_{tr}\mathbf W]\_i-[\mathbf M\_i\mathbf C\mathbf Y\_{tr}\mathbf W]\_i)\\\\
> &=
> \frac{1}{n}\sum\_{i\in D\_{tr}}\ell(\mathbf y\_i,[\mathbf P\mathbf Y\_{tr}\mathbf W]\_i-[\mathbf C\mathbf Y\_{tr}\mathbf W]\_i)\\\\
> &=
> \frac{1}{n}\sum\_{i\in D\_{tr}}\ell(\mathbf y\_i,[(\mathbf P-\mathbf C)\mathbf Y\_{tr}\mathbf W]\_i).
> \end{aligned}
> $$
>
> **Question**: More generally, the paper is at times hard to read due to it being notationally dense (particularly the excessive use of subscripts), and the stream of the 'Remarks'. I feel the paper could benefit from one more pass of re-writing.
>
> **Response:** In our revised version, we have made the notation/presentation more clear and moved one of the less critical remarks to the supplementary.

---

> > ### Comment · Reviewer_bkGb · 2021-11-26
> > **Thank you for your response**
> >
> > To clarify, I believe the paper's main contribution is the analysis of the stochastic label trick in a linearized GNN, and revealing it as a sum of deterministic loss terms. In contrast, the newly-added Theorem 3, which is claimed to be an analysis of GNNs, really just applies to label-dependent functions $f$ with bounded loss and arises from simpler facts of conditional expectation and the binomial distribution (please correct me if I'm wrong). In particular it is not specific to graph-based models and does not arise as a consequence of spreading features/labels on the graph (since it does not involve the parameterization of the adjacency/propagation matrix). That said, I believe that the analysis of Theorem 1 and Corollary 1 themselves are strong and provide useful insight of the label trick procedure.
> >
> > Furthermore, as the authors admit, Theorem 3 is limited in scope: "... produce different predictions for all test nodes depending on $i$ unless $f_{GNN}$ is linear." Whereas one of the claimed primary contributions of the paper was "a data-fitting term that naturally resolves potential label leakage issues and _maintains consistent predictions_ during training and inference"
> >
> > For regression task, I re-emphasize that $R^2$ should be used for comparison in the regression setting, regardless of what the previous paper used; if a constant function fits the data well then $R^2$ will be low.
> >
> > I'm also curious how the running time compares for trainable and non-trainable Label Propagation?
> >
> > If the regularizer was not used in Table 3, then are there any results showing its effectiveness? It should be possible for the regression datasets in Table 2, correct? The decomposition of the stochastic label trick does not seem to be corroborated by experiments in the paper then. This seems to be a major gap between the derived theoretical results and the experiments presented.
> >
> > Finally, I know the authors claimed that the right figure showing the deterministic label trick on GNNs is prohibitively expensive to compute on the ogbn-arxiv dataset, but it would be good to see its comparison on a smaller dataset at least in the final revision.

---

> > > ### Author Response · Authors · 2021-11-28
> > > **Response to Reviewer bkGb Part 2**
> > >
> > > **Question:** I'm also curious how the running time compares for trainable and non-trainable Label Propagation?
> > >
> > > **Response:**
> > > For inference, trainable label propagation and regular label propagation require essentially the same amount of time (i.e., the weight matrix multiplication for the trainable version introduces negligible complexity). But of course regular label propagation does not require training.  Even so, relative to typical GNN models, trainable label propagation is substantially more efficient, in large part because expensive feature propagation is not required.  For example, the following table shows comparative results between trainable LP and a 3-layer GCN on ogbn-arxiv.
> > >
> > > |Model|Training time (ogbn-arxiv)|
> > > |:-|:-:|
> > > |Trainable Label Propagation|270s|
> > > |3-layer GCN|3419s|
> > >
> > > In this way trainable LP provides a fast simple baseline that is naturally scalable.
> > >
> > > **Question:** If the regularizer was not used in Table 3, then are there any results showing its effectiveness? It should be possible for the regression datasets in Table 2, correct? The decomposition of the stochastic label trick does not seem to be corroborated by experiments in the paper then. This seems to be a major gap between the derived theoretical results and the experiments presented.
> > >
> > > **Response:** Very interesting suggestion.  Note though that the regression experiments from Table 2 involve tabular data and a gradient-boosted decision tree (GBDT) base model whose parameterization does not directly align with the weight-based regularization of Theorem 1 and Corollary 1.  Moreover, in order to respond promptly before the discussion period ends, we do not have time to search for and experiment with suitable alternative regression datasets.  However, we can quickly apply the theoretically-derived regularization factor to classification experiments involving the deterministic self-excluded linear predictors from Section 4.2.  The table below provides such an example using the Cora dataset and the TWIRLS linear predictor plus regularization as $\alpha$ is varied. (Note that if $\alpha = 0$, there is infinite penalty on the label weights which is equivalent to no labels; in contrast if $\alpha = 1$ the regularization factor is zero.)
> > >
> > > | $\alpha$ | label trick (D)|
> > > |-|:-:|
> > > |0.0 (no labels)|82.41|
> > > |0.1|83.76|
> > > |0.2|83.89|
> > > |0.3|83.89|
> > > |0.4|84.01|
> > > |0.5|84.01|
> > > |0.6|84.26|
> > > |0.7|84.38|
> > > |0.8|84.26|
> > > |0.9|84.13|
> > > |1.0 (no regularization)|83.89|
> > >
> > > From these results we observe that including labels plus the theoretically-motivated regularization (with the right value for $\alpha$) can indeed improve performance.  This suggests that even with a classification loss (which only loosely aligns with the stated theorem/corollary assumption of an MSE loss), the theory has some practical/predictive value in producing a useful regularization factor.  We can include these results and further related experiments in the revision.
> > >
> > > **Question:** Finally, I know the authors claimed that the right figure showing the deterministic label trick on GNNs is prohibitively expensive to compute on the ogbn-arxiv dataset, but it would be good to see its comparison on a smaller dataset at least in the final revision.
> > >
> > > **Response:** Good suggestion; we can add this in the revision (note that the reviewer's proposal cannot be efficiently implemented by straightforward modifications to standard GNN codebases; however, we can still do this given sufficient time for the revision).

---

> > > ### Author Response · Authors · 2021-11-28
> > > **Response to Reviewer bkGb Part 1**
> > >
> > > Thanks for the great/insightful follow-up questions (these will definitely help to improve the current draft). We address each one below.
> > >
> > > **Question:** To clarify, I believe the paper's main contribution is the analysis of the stochastic label trick in a linearized GNN, and revealing it as a sum of deterministic loss terms. In contrast, the newly-added Theorem 3, which is claimed to be an analysis of GNNs, really just applies to label-dependent functions $f$ with bounded loss and arises from simpler facts of conditional expectation and the binomial distribution (please correct me if I'm wrong). In particular it is not specific to graph-based models and does not arise as a consequence of spreading features/labels on the graph (since it does not involve the parameterization of the adjacency/propagation matrix). That said, I believe that the analysis of Theorem 1 and Corollary 1 themselves are strong and provide useful insight of the label trick procedure.
> > >
> > > **Response:**  We agree that Theorem 3 is not the primary contribution, and the reviewer correctly points out that Theorem 1 and Corollary 1 are the stronger contributions.  That being said, Theorem 3 is nonetheless still quite relevant for graph-based models whereby the prediction for a given node is dependent on other nodes.  If this were not the case (i.e., all graph edges are removed), then the equivalent of Theorem 3 would actually hold trivially regardless of the value of $\alpha$, since there is no influence from other samples.  The point then of this result is to show that even with arbitrary dependency structure between nodes/samples (whether instantiated via a GNN or some other architecture), the $\alpha \rightarrow 1$ limit shares some properties with the linear case.  So in this sense Theorem 3 is only meaningful/non-trivial for graph-based models, and trivially true otherwise.
> > >
> > > **Question:** Furthermore, as the authors admit, Theorem 3 is limited in scope: "... produce different predictions for all test nodes depending on $i$ unless $f_{GNN}$ is linear." Whereas one of the claimed primary contributions of the paper was "a data-fitting term that naturally resolves potential label leakage issues and *maintains consistent predictions* during training and inference"
> > >
> > > **Response:**  Our main purpose is to closely analyze the label trick, and show potential areas of both strength and potentially weakness where appropriate.  As noted in the introduction and elsewhere, we do state that *under certain restrictive settings* the label trick resolves potential label leakage issues and maintains consistent predictions during training and inference.  However, this need not always be the case, and as the discussion below Theorem 3 elucidates, for a nonlinear GNN we can no longer strictly guarantee that predictions will be exactly consistent between training and testing.  Overall, we believe that raising this distinction, i.e. detailing the special case where consistency holds and a more general scenario where it may not, is actually an important analysis point in and of itself which has not been previously addressed or even acknowledged in the literature.  Anyway, thanks for bringin up this issue, we can certainly clarify in a revision.
> > >
> > > **Question:** For regression task, I re-emphasize that $R^2$ should be used for comparison in the regression setting, regardless of what the previous paper used; if a constant function fits the data well then $R^2$ will be low.
> > >
> > > **Response:** Below we convert the results from Table 2 in our submission to their corresponding $R^2$ values, and we can include these numbers in a revised version of the paper.
> > >
> > > |    Dataset        |  C&S    |  Trainable C&S |
> > > |:-----------------|:-------:|:------:|
> > > |   House        | 0.797 ± 0.005 | **0.854 ± 0.005**   |
> > > |  County | 0.854 ± 0.005  | **0.788 ± 0.015**   |
> > > | VK   | 0.163 ± 0.011  | **0.172 ± 0.011**   |
> > > | Avazu    |  0.405 ± 0.036  | **0.413 ± 0.033**   |
> > >
> > > Note that the $R^2$ values on the VK dataset are somewhat low; however, this is a particularly challenging dataset and our results are consistent with the current SOTA performance.  Specifically, the BGNN model from reference (Boost then convolve: Gradient boosting meets graph neural networks by Ivanov & Prokhorenkova) reports an MSE of 6.95±0.21 for the VK dataset.  While $R^2$ values are not reported in this paper, given that Trainable C&S achieves 6.95±0.22 MSE, the corresponding $R^2$ values should be nearly the same (note also that the training protocols were identical).

---

> > > > ### Comment · Reviewer_bkGb · 2021-11-29
> > > > **Thank you for the response**
> > > >
> > > > Thank you for your response. I think it is worth highlighting that after looking at the $R^2$ score, that the House dataset is the only one with meaningful improvement (and I am not sure why the right-column is bolded for the County dataset). This is not to fault the experiment/methodology, though.
> > > >
> > > > I find the results of the regularization effect very interesting! I would like to thank the authors once more for all the experiments they include in their response.
> > > >
> > > > As a final remark, I believe the ideas of this paper are very strong, and that the contributions are insightful and novel. However, even after reviewing period, I do think that there remains non-trivial changes for the final revision. For this reason, I am keeping my score.

---

> > > > > ### Author Response · Authors · 2021-11-30
> > > > > **Final Response and Important Correction**
> > > > >
> > > > > **Question:** I think it is worth highlighting that after looking at the $R^2$ score, that the House dataset is the only one with meaningful improvement (and I am not sure why the right-column is bolded for the County dataset). This is not to fault the experiment/methodology, though.
> > > > >
> > > > > **Response:**  Actually, in the previous table we accidentally recorded the House datset Trainable C&S result in the County C&S slot (notice the identical numbers); the correct table is included below.  From this we observe that Trainable C&S has a higher $R^2$ value in all cases.
> > > > >
> > > > > |    Dataset        |  C&S    |  Trainable C&S |
> > > > > |:-----------------|:-------:|:------:|
> > > > > |   House        | 0.797 ± 0.005 | **0.854 ± 0.005**   |
> > > > > |  County | 0.625 ± 0.048  | **0.788 ± 0.015**   |
> > > > > | VK   | 0.163 ± 0.011  | **0.172 ± 0.011**   |
> > > > > | Avazu    |  0.405 ± 0.036  | **0.413 ± 0.033**   |
> > > > >
> > > > > Note that with the exception of Avazu, all improvements are statistically significant once we account for the fact that the variability is largely due to the randomness introduced by different splits, such that significant differences can be maintained even with higher standard errors (see Appendix A.2 for further explanation).  For example, on the VK dataset, across all trials Trainable C&S can achieve a higher $R^2$ value than C&S; see expanded table below with trial-to-trial results.  Hence in aggregate, the end-to-end Trainable C&S really does exhibit a meaningful improvement.
> > > > >
> > > > > |        |               | seed 1         | seed 2         | seed 3         | seed 4         | seed 5         |
> > > > > |--------|---------------|----------------|----------------|----------------|----------------|----------------|
> > > > > | House  | Trainable C&S | **0.8527**   | **0.8456**    | **0.8660** | **0.8480** | **0.8600** |
> > > > > |        | C&S           | 0.8016     | 0.7867      | 0.7984      | 0.7947      | 0.8015     |
> > > > > | County | Trainable C&S | **0.7770**   | **0.7702** | **0.8032**  | **0.8082**  | **0.7836**  |
> > > > > |        | C&S           | 0.6255     | 0.6392     | 0.5371     | 0.6844     | 0.6394      |
> > > > > | VK     | Trainable C&S | **0.1801** | **0.1597** | **0.1604** | **0.1734** | **0.1878** |
> > > > > |        | C&S           | 0.1737     | 0.1481      | 0.1535     | 0.1611     | 0.1769     |
> > > > > | Avazu  | Trainable C&S | 0.4167      | **0.4628** | **0.4319** | **0.3742**  | **0.3780** |
> > > > > |        | C&S           | **0.4189** | 0.4511      | 0.4280     | 0.3649     | 0.3608     |
> > > > >
> > > > > Note also, that although the $R^2$ values are relatively low for Avazu and VK datasets, as mentioned in a previous response, this difficulty is shared by SOTA GNN-based models, e.g., BGNN.
> > > > >
> > > > > **Question:** I find the results of the regularization effect very interesting! I would like to thank the authors once more for all the experiments they include in their response.
> > > > >
> > > > > **Response:** This can be included in the revision, good suggestion.
> > > > >
> > > > > **Question:** As a final remark, I believe the ideas of this paper are very strong, and that the contributions are insightful and novel. However, even after reviewing period, I do think that there remains non-trivial changes for the final revision. For this reason, I am keeping my score.
> > > > >
> > > > > **Response:** Thanks for the reviewer's appreciation of the novelty of our paper. However, given that the empirical results mentioned above are actually significant once our transcription error is corrected, we hope that this addresses the last non-trivial change/comment that might be needed (with the assumption that discussion points from OpenReview will be incorporated into the revision as well).

---

> ### Author Response · Authors · 2021-11-18
> **Response to Reviewer bkGb Part 2**
>
> **Question**: For Table 2, $R^2$ value should be used to evaluate regression.
>
> **Response:** It is true that the $R^2$ metric is a quite suitable choice for regression as it is independent of the prediction scale. However, both the regression datasets and the use of mean square error loss we adopted were inherited from "Boost then convolve: Gradient boosting meets graph neural networks" by Ivanov & Prokhorenkova as described in the paper.  Regardless, if the reviewer prefers, for the final version we can rerun and switch to $R^2$ values.
>
> **Question**: Section 4.1 is unclear and should have greater detail. For example, how is this procedure trained? What is the final model (is it just supervision on the Neumann series with a weight matrix instead of a propagation matrix)? Further, some reference should be given to other papers who learn/modify the adjacency/propagation matrix.
>
> **Response:** We used cross entropy as the loss function and trained the label trick as described in Definition 1, using the Neumann series as $\mathbf P$ for consistency with standard label propagation. Further hyperparameter details can be found in the supplementary. We have also further clarified Section 4.1 and added more supporting references.
>
> **Question:** Why isn't the deterministic label trick used for comparison in the right plot in Figure 1? Further, which GNN is used? I believe this is unspecified in Section 5.3.
>
> **Response:** For nonlinear GNNs, a deterministic self-excluded version of the label trick is in fact possible in the limit $\alpha \rightarrow 1$ per our new Theorem 3.  However, because the r.h.s. of equation (12) in the revision does not further simplify as in the linear case, the self-excluded version with GNNs can be computationally prohibitive to execute, which is why it was excluded from Figure 1 (right).  Regarding which GNN is used, although previously we had deferred the implementation details to the supplementary, we have now specified in the main text that we used a regular three-layer GCN in Section 5.3.
>
> **Question:** In remark 4, if the regularization is shut off why is there "no chance for overfitting"?
>
> **Response:**  In remark 4, we do not state that shutting off regularization implies that there is no chance for overfitting.  Rather, remark 4 mentions that if the graph has no edges, then "there is no chance for overfitting to the labels."  This is a direct consequence of using the label trick, since without edges, it is impossible to overfit to neighboring labels, i.e., all nodes are isolated so if a given node is in $\mathcal{D}_{out}$, it has no input label nor any connection to a neighbor with an input label.
>
> **Question:** If the derivation shows that we can eventually decompose with a regularization term in the deterministic case, is this used for results in Table 3? If it is not used (as it does not seem to be mentioned), why not?
>
> **Response:** Table 3 involves testing the deterministic simplification of the label trick as applied to the non-linear GNN models with linear propagation layers presented in Section 4.2. This is described in the text of Section 5.2.  Moreover, no additional regularization is involved as the presented cases correspond with the basic self-excluded, $\alpha \rightarrow 1$ limit.

---

> ### Author Response · Authors · 2021-11-26
> **To Reviewer bkGb**
>
> We hope our comments have adequately addressed the reviewer's previous concerns. Given the limited remaining time of the rebuttal session, we wonder if the reviewer has any more feedback?

---

### Official Review · Reviewer_xaW4 · 2021-11-01

**Correctness:** 3
**Technical Novelty And Significance:** 2
**Empirical Novelty And Significance:** 2
**Recommendation:** 5
**Confidence:** 3

**Main Review:**

Strong points:
1.	The paper provides a relative new framework for GNNs where the stochastic label trick is replaced by a deterministic alternative which focuses on exclusion of self propagation, the alternative seems to be novel
2.	Showing an upper bound in  Thm 2 on the deterministic proposal with respect to the cross entropy loss is a valuable result.

Weak points:
1.	Motivation: I still feel unconvinced on what are the reasons for replacing the non-deterministic label trick with a deterministic one, except of course under sampling. Is there a computation bottleneck as well?
2.	The authors claim that prior use of the label trick has not been theoretically substantiated but what is the Deterministic advantage here in terms accuracy/ speed/ any other advantage?
3.	The experimental validation leaves me a bit puzzled: except for the graph in Fig 1 there no actual comparison between the deterministic and SotA random label tricks. The authors instead focus on comparisons that are not directly related to what they propose in the paper (e.g. trainable vs non-trainable, with label trick vs without). Why is that so?
4.	My main concern is that the experimental validation even for the experiments on the trainable version and use of the label trick is not convincing enough. I see that in some examples the difference between either the trainable label propagation vs the traditional one, as well as the use of the label trick do not show significant improvement (if at all) in particular when the STD is taken into account.
5.	There are some missing important references on the subject of label propagation in graph and the incorporation of label data:
a.	Regularization on Graphs with Function-adapted Diffusion Processes Arthur D. Szlam, Mauro Maggioni, Ronald R. Coifman; 9(55):1711−1739, 2008.
6.	Many of the algorithms mentioned are not fully understood and self-contained to be easily addressed by the reader, also what is the reference to C&S?


**Summary Of The Paper:**

The paper proposes a deterministic version for the label trick in GNNs (and other graph frameworks), that is based on a self-exclusion operator for the propagation process in order to avoid label leakage. The authors propose some applications in other label propagation settings. Experiments are shown to compare with regular label propagation, with and without label trick, effect of regularization term in the deterministic setting vs randomized one.

**Summary Of The Review:**

Some fundamental issues in the paper are preventing me from accepting this paper: the experimental validation is insufficient and unconvincing. The motivation for proposing the deterministic alternative to the label trick is not discussed.

---

> ### Author Response · Authors · 2021-11-18
> **Response to Reviewer xaW4 Part 1**
>
> Thanks for the detailed comments. We have now posted a new version of the paper which resolves some of the issues that were raised regarding the original submission, and includes one new technical result (Theorem 3).  Below we also respond point-by-point to each reviewer comment.
>
> **Question:** Motivation: I still feel unconvinced on what are the reasons for replacing the non-deterministic label trick with a deterministic one, except of course under sampling. Is there a computation bottleneck as well?
>
> **Response:** Just to be clear, we are *not* suggesting that the original, stochastic label trick should be replaced with the deterministic version.  Rather, the latter is primarily introduced in Section 3 as an analysis tool to help better understand the behavior of the stochastic version.  That being said, if $\alpha \rightarrow 1$, the stochastic label trick asymptotically approaches the deterministic (self-excluded) version as described in Section 3.3 and the new Section 3.4 (which provides a general nonlinear extension via Theorem 3).  Consequently, as discussed in Section 5, in this particular situation the deterministic version could be preferable since it can speed up the training process by avoiding sampling altogether.
>
> **Question**: The authors claim that prior use of the label trick has not been theoretically substantiated but what is the Deterministic advantage here in terms accuracy/ speed/ any other advantage?
>
> **Response:**  As mentioned in Section 1, one of our main contributions is the demonstration of how the original stochastic label trick can be reduced to a more interpretable, deterministic training objective that clearly elucidates expected behaviors.  However, similar to our response above, this deterministic form motivates an alternative, non-random version of the label trick that can be practically-relevant in special circumstances.  In particular, when the splitting probability $\alpha$ tends to one, the self-excluded deterministic label trick closely approximates the original stochastic version, but is more stable and efficient to train (e.g., in the experiments for Figure 1, when $\alpha>0.9$ the stochastic version takes 5-100 times more epochs to converge than the deterministic one).  But of course for broader choices of $\alpha$ and general architectures the stochastic version must still be relied upon.
>
> **Question:** The experimental validation leaves me a bit puzzled: except for the graph in Fig 1 there no actual comparison between the deterministic and SotA random label tricks. The authors instead focus on comparisons that are not directly related to what they propose in the paper (e.g. trainable vs non-trainable, with label trick vs without). Why is that so?
>
> **Response:**  Figure 1 shows that indeed, as $\alpha$ approaches one, the stochastic label trick approaches the deterministic self-excluded one as expected.  Hence in $\alpha \approx 1$ regimes we can safely rely on the simpler deterministic version (at least for linear cases; for complex GNNs the deterministic version can still potentially be very expensive to compute).  However, we do not suggest that the deterministic label trick is generally superior, and most of our experiments are designed to show how *new use-cases* of the original stochastic label trick compare against baselines without any label trick.
>
> **Question:** My main concern is that the experimental validation even for the experiments on the trainable version and use of the label trick is not convincing enough. I see that in some examples the difference between either the trainable label propagation vs the traditional one, as well as the use of the label trick do not show significant improvement (if at all) in particular when the STD is taken into account.
>
> **Response:** The reviewer has questioned the significance of the results in Tables 1-4.  We address these results table by table.
>
> *Table 1*: The label trick already demonstrates statistically significant results on both Arxiv and Products in Table 1. Moreover, the Cora-full results are also significant once we account for the fact that the higher variance mainly comes from the random division of the dataset into training, validation and test dataset.  In this way, the performance gap remains consistent across trials such that the improvement is actually still significant. For example, the following table shows that the label trick consistently improves the results in Table 1 on Cora-full over different dataset splits even after careful tuning of hyperparameters on the baseline model.  Hence it is only on Pubmed where only a small, possibly negligible advantage is displayed.

---

> > ### Comment · Reviewer_xaW4 · 2021-11-20
> > **Thank you for your response**
> >
> > I better understand your motivation. And I thank you for the additional results and explanation. As an overall contribution I think that showing that the label trick works better in your experimental validation is OK, but I consider that as prior knowledge. On the other hand, the fact that the randomized trick can be done deterministically with more stability/accuracy/any other advantage, is something I consider as of higher importance and novelty. But that is not shown to this end. Hence Im in agreement with reviewers' Ugm6 points, on others, which also see the experimental validation problematic on a similar basis.

---

> > > ### Author Response · Authors · 2021-11-21
> > > **Response to reviewer xaW4**
> > >
> > > Just to clarify, our empirical results all represent new use cases of the label trick (including new usage of the original stochastic version, e.g., Figures 1 and 2), not prior knowledge or reproduction of prior results.  That being said, our primary contribution is *not* any of these empirical comparisons.  Rather, our main contribution is the new interpretation of the original stochastic label trick afforded by our analysis in Section 3.  This is critical given that the intrinsic randomness cannot otherwise be well-explained considering the impact of graph structure and exponential number of possible splits.  Moreover, the impact of the splitting probability, and its explicit relationship to regularization, was also previously unexplained.  Within this context, the decomposition into the self-excluded loss plus adaptive regularization we have derived directly elucidates these issues.

---

> > > > ### Comment · Reviewer_xaW4 · 2021-11-26
> > > > **Thank you for all clarifications. I have changed my score now to reflect this**
> > > >
> > > > Thank you. You have clarified the significance of some of your contributions. I have updated my score accordingly.

---

> ### Author Response · Authors · 2021-11-18
> **Response to Reviewer xaW4 Part 2**
>
> Table: Accuracy results on Cora-full.
>
> |  Trial  | Label Propagation | Trainable Label Propagation |
> |:-------:|:-----------------:|:---------------------------:|
> |    1    |       65.52       |          **67.57**          |
> |    2    |       65.77       |          **66.03**          |
> |    3    |       64.84       |          **66.28**          |
> |    4    |       66.00       |          **66.23**          |
> |    5    |       68.10       |          **68.58**          |
> |    6    |       67.39       |          **67.87**          |
> |    7    |       67.29       |          **68.07**          |
> |    8    |       66.86       |          **68.83**          |
> |    9    |       66.48       |          **67.85**          |
> |   10    |       66.13       |          **66.73**          |
> | Average |   66.44 ± 0.93    |      **67.40 ± 0.96**       |
>
> *Table 2*: All improvements except for the Avazu dataset are significant once we account for shared, trial-to-trial variability (in the same fashion as we showed for Cora-full above).
>
> *Table 3*:  These results are meant to highlight broader use cases of the deterministic simplification of the label trick.  And the Arxiv results do show significant improvement in Table 3; however, for both Cora-full and Pubmed the performances with and without the label trick are indeed similar.  As mentioned in the paper (end of Section 5.2), this is because for these datasets under the current experimental settings, the training accuracy is near 100% such that the label information is already adequately embedded in the model and the label trick has limited space for impact.
>
> To further illustrate the role of training accuracy in training with the label trick, we produce two new datasets by projecting the input features on Cora-full and Pubmed to 32 dimensions via PCA. This is a much more challenging task because less feature information is available for prediction. We train SGC, SIGN and TWIRLS under the same settings as in Table 3. The following tables show the test and training accuracy on these datasets respectively, as well as results using original features as in Table 3.
>
> Table: Test accuracy with/without label trick (D).
>
> |       Dataset       |       SGC        |       SGC        |       SIGN       |       SIGN       |    TWIRLS    |      TWIRLS      |
> |:-------------------:|:----------------:|:----------------:|:----------------:|:----------------:|:------------:|:----------------:|
> | **label trick (D)** |        ✗         |        ✓         |        ✗         |        ✓         |      ✗       |        ✓         |
> |    Cora-full-32d    |   55.63 ± 0.53   | **62.37 ± 0.87** |   55.10 ± 0.55   | **62.05 ± 0.47** | 58.57 ± 0.48 | **61.74 ± 0.46** |
> |     Pubmed-32d      |   83.15 ± 0.41   | **84.33 ± 0.68** |   84.75 ± 0.37   | **85.72 ± 0.84** | 84.94 ± 0.37 | **87.60 ± 0.44** |
> |      Cora-full      | **65.87 ± 0.61** |   65.81 ± 0.69   | **68.54 ± 0.76** |   68.44 ± 0.88   | 70.36 ± 0.51 | **70.40 ± 0.71** |
> |       Pubmed        |   85.02 ± 0.43   | **85.23 ± 0.57** |   87.94 ± 0.52   | **88.09 ± 0.59** | 89.81 ± 0.56 | **90.08 ± 0.52** |
>
> Table: Training accuracy with/without label trick (D).
>
> |       Dataset       |     SGC      |       SGC        |     SIGN     |       SIGN       |      TWIRLS      |      TWIRLS      |
> |:-------------------:|:------------:|:----------------:|:------------:|:----------------:|:----------------:|:----------------:|
> | **label trick (D)** |      ✗       |        ✓         |      ✗       |        ✓         |        ✗         |        ✓         |
> |    Cora-full-32d    | 57.69 ± 0.21 | **65.75 ± 0.19** | 57.20 ± 0.21 | **65.88 ± 0.27** |   63.44 ± 0.41   | **67.40 ± 0.36** |
> |     Pubmed-32d      | 83.43 ± 0.35 | **84.74 ± 0.17** | 84.96 ± 0.13 | **86.05 ± 0.19** |   85.97 ± 0.22   | **88.43 ± 0.26** |
> |      Cora-full      | 95.31 ± 0.16 | **97.59 ± 0.12** | 99.78 ± 0.03 | **99.99 ± 0.01** | **99.95 ± 0.01** |   99.76 ± 0.05   |
> |       Pubmed        | 85.63 ± 0.17 | **85.69 ± 0.23** | 91.26 ± 0.12 | **91.43 ± 0.17** |   97.32 ± 0.30   | **97.48 ± 0.28** |
>
> From the two tables above, we can see that the degree to which the label trick can improve accuracy is directly related to the training accuracy. When the training accuracy is near 100%, the label trick could hardly benefit the model as the label information is already adequately embedded in the model. However, when the training accuracy is lower, the label trick can increase both the training and test accuracy significantly (once the variance from the dataset divisions is taken into account).
>
> Moreover, for large-scale graphs it may be less likely that high accuracy or overfitting is obtained during training due to the large number of training samples.  So in such cases, the label trick is more likely to be effective.

---

> ### Author Response · Authors · 2021-11-18
> **Response to Reviewer xaW4 Part 3**
>
> *Table 4*: Although a few results are comparable to those without the label trick, our main goal is to present a novel use case and to show that C&S is parameterizable. However, we have only used the simplest linear model here, which has been able to achieve decent results. This motivates its application to more complex cases, including nonlinear models, where regular C&S cannot perform.
>
> Regardless, the most important contribution of our paper is not promoting a new technique per se, but rather, to provide a theoretical foundation for the label trick, while demonstrating some previously-untried applications of the label trick that can motivate further use cases.
>
> **Question:** There are some missing important references on the subject of label propagation in graph and the incorporation of label data: a. Regularization on Graphs with Function-adapted Diffusion Processes Arthur D. Szlam, Mauro Maggioni, Ronald R. Coifman; 9(55):1711−1739, 2008.
>
> **Response:**  Thanks for the suggestion.  While this paper presents many interesting ideas, it is somewhat orthogonal to the message of our submission.  In particular, this reference describes a process for extracting graphs from data based on various similarity metrics and then applying function-space regularization using various diffusion processes. But there is no reference to existing work on label propagation nor any head-to-head comparisons.  That being said, perhaps these types of diffusion processes could be adopted to motivate a richer family of label propagation algorithms; however, this is beyond the scope of our present work.
>
> **Question**: Many of the algorithms mentioned are not fully understood and self-contained to be easily addressed by the reader, also what is the reference to C&S?
>
> **Response:** Previously we cited C&S in Section 1 and subsequently only referred to the name, but now in the revised version we have included this reference in multiple locations for clarity; good suggestion.  As for other algorithms, there is unfortunately limited space for full details; however, hopefully the revision is generally more clear with adequate citations where needed.

---

### Official Review · Reviewer_Ugm6 · 2021-11-02

**Correctness:** 3
**Technical Novelty And Significance:** 3
**Empirical Novelty And Significance:** 1
**Recommendation:** 6
**Confidence:** 4

**Details Of Ethics Concerns:**

None.

**Main Review:**

(1) The effectiveness of the label trick in improving GNN performance has already been demonstrated in prior work. Because of that, this paper does not repeat these efforts and instead tries to showcase the broader application scenarios from Sec 4. I understand this goal. However, results in the four application scenarios that the authors selected indicate little or no benefit of the label trick. In particular, closely examining, Sec 5.1-5.4 and Tables 1-4, I see that using the label trick gives performance comparable to the baseline methods. This happens in many (if not most) scenarios when I take into account the standard deviation across ten independent runs.

(2) Following from the previous point, I would like to get more clarity on the role of label trick in GNN training. The study makes a very shallow remark that: "For example, at the time of this submission, the top 10 results spanning multiple research teams all rely on the label trick, as do the top 3 results from the recent KDDCUP 2021 Large-Scale Challenge MAG240M-LSC." That is okay, however, it does not mean that the label trick necessarily leads to better performance. To be able to argue that, one would need to run an ablation study comparing top models from the Challenge with and without the label trick. Are there other desirable effects that made the label trick so popular (e.g., faster convergence, more robust training etc.)?

**Summary Of The Paper:**

This paper examines the "label trick" which enables the parallel propagation of labels and features and benefits various SOTA GNN architectures. The label trick has not yet been rigorously analyzed and so this paper studies it from a theoretical perspective. It first introduces a self-excluded simplification of the label trick by applying an extreme splitting strategy. It then shows that the random splits of the label trick can be regarded as regularization of self-excluded label weights. The paper also discusses broader applications of the label trick in order a) to empower graph-based methods with trainable weights and b) to get rid of the randomness effect by incorporating self-excluded propagation within GNNs composed of linear propagation layers.

**Summary Of The Review:**

I appreciate the formal description of the label trick and the associated theoretical analysis. However, the novel use-cases motivated by that analysis are weak and empirical analysis on the role of the label trick is incomplete.

---

> ### Author Response · Authors · 2021-11-18
> **Response to Reviewer Ugm6 Part 1**
>
> Thanks for the detailed comments. We have now posted a new version of the paper which resolves some of the issues that were raised regarding the original submission, and includes one new technical result (Theorem 3). Below we also respond to each reviewer comment.
>
> **Question:** (1) The effectiveness of the label trick in improving GNN performance has already been demonstrated in prior work. Because of that, this paper does not repeat these efforts and instead tries to showcase the broader application scenarios from Sec 4. I understand this goal. However, results in the four application scenarios that the authors selected indicate little or no benefit of the label trick. In particular, closely examining, Sec 5.1-5.4 and Tables 1-4, I see that using the label trick gives performance comparable to the baseline methods. This happens in many (if not most) scenarios when I take into account the standard deviation across ten independent runs.
>
> **Response:** The reviewer has questioned the significance of the results in Tables 1-4.  We address these results table by table.
>
> *Table 1*: The label trick already demonstrates statistically significant results on both Arxiv and Products in Table 1. Moreover, the Cora-full results are also significant once we account for the fact that the higher variance mainly comes from the random division of the dataset into training, validation and test dataset.  In this way, the performance gap remains consistent across trials such that the improvement is actually still significant. For example, the following table shows that the label trick consistently improves the results in Table 1 on Cora-full over different dataset splits even after careful tuning of hyperparameters on the baseline model.  Hence it is only on Pubmed where only a small, possibly negligible advantage is displayed.
>
> Table: Accuracy results on Cora-full.
>
> |  Trial  | Label Propagation | Trainable Label Propagation |
> |:-------:|:-----------------:|:---------------------------:|
> |    1    |       65.52       |          **67.57**          |
> |    2    |       65.77       |          **66.03**          |
> |    3    |       64.84       |          **66.28**          |
> |    4    |       66.00       |          **66.23**          |
> |    5    |       68.10       |          **68.58**          |
> |    6    |       67.39       |          **67.87**          |
> |    7    |       67.29       |          **68.07**          |
> |    8    |       66.86       |          **68.83**          |
> |    9    |       66.48       |          **67.85**          |
> |   10    |       66.13       |          **66.73**          |
> | Average |   66.44 ± 0.93    |      **67.40 ± 0.96**       |
>
> *Table 2*: All improvements except for the Avazu dataset are significant once we account for shared, trial-to-trial variability (in the same fashion as we showed for Cora-full above).
>
> *Table 3*:  These results are meant to highlight broader use cases of the deterministic simplification of the label trick.  And the Arxiv results do show significant improvement in Table 3; however, for both Cora-full and Pubmed the performances with and without the label trick are indeed similar.  As mentioned in the paper (end of Section 5.2), this is because for these datasets under the current experimental settings, the training accuracy is near 100% such that the label information is already adequately embedded in the model and the label trick has limited space for impact.
>
> To further illustrate the role of training accuracy in training with the label trick, we produce two new datasets by projecting the input features on Cora-full and Pubmed to 32 dimensions via PCA. This is a much more challenging task because less feature information is available for prediction. We train SGC, SIGN and TWIRLS under the same settings as in Table 3. The following tables show the test and training accuracy on these datasets respectively, as well as results using original features as in Table 3.
>
> Table: Test accuracy with/without label trick (D).
>
> |       Dataset       |       SGC        |       SGC        |       SIGN       |       SIGN       |    TWIRLS    |      TWIRLS      |
> |:-------------------:|:----------------:|:----------------:|:----------------:|:----------------:|:------------:|:----------------:|
> | **label trick (D)** |        ✗         |        ✓         |        ✗         |        ✓         |      ✗       |        ✓         |
> |    Cora-full-32d    |   55.63 ± 0.53   | **62.37 ± 0.87** |   55.10 ± 0.55   | **62.05 ± 0.47** | 58.57 ± 0.48 | **61.74 ± 0.46** |
> |     Pubmed-32d      |   83.15 ± 0.41   | **84.33 ± 0.68** |   84.75 ± 0.37   | **85.72 ± 0.84** | 84.94 ± 0.37 | **87.60 ± 0.44** |
> |      Cora-full      | **65.87 ± 0.61** |   65.81 ± 0.69   | **68.54 ± 0.76** |   68.44 ± 0.88   | 70.36 ± 0.51 | **70.40 ± 0.71** |
> |       Pubmed        |   85.02 ± 0.43   | **85.23 ± 0.57** |   87.94 ± 0.52   | **88.09 ± 0.59** | 89.81 ± 0.56 | **90.08 ± 0.52** |

---

> ### Author Response · Authors · 2021-11-18
> **Response to Reviewer Ugm6 Part 2**
>
>
> Table: Training accuracy with/without label trick (D).
>
> |       Dataset       |     SGC      |       SGC        |     SIGN     |       SIGN       |      TWIRLS      |      TWIRLS      |
> |:-------------------:|:------------:|:----------------:|:------------:|:----------------:|:----------------:|:----------------:|
> | **label trick (D)** |      ✗       |        ✓         |      ✗       |        ✓         |        ✗         |        ✓         |
> |    Cora-full-32d    | 57.69 ± 0.21 | **65.75 ± 0.19** | 57.20 ± 0.21 | **65.88 ± 0.27** |   63.44 ± 0.41   | **67.40 ± 0.36** |
> |     Pubmed-32d      | 83.43 ± 0.35 | **84.74 ± 0.17** | 84.96 ± 0.13 | **86.05 ± 0.19** |   85.97 ± 0.22   | **88.43 ± 0.26** |
> |      Cora-full      | 95.31 ± 0.16 | **97.59 ± 0.12** | 99.78 ± 0.03 | **99.99 ± 0.01** | **99.95 ± 0.01** |   99.76 ± 0.05   |
> |       Pubmed        | 85.63 ± 0.17 | **85.69 ± 0.23** | 91.26 ± 0.12 | **91.43 ± 0.17** |   97.32 ± 0.30   | **97.48 ± 0.28** |
>
> From the two tables above, we can see that the degree to which the label trick can improve accuracy is directly related to the training accuracy. When the training accuracy is near 100%, the label trick could hardly benefit the model as the label information is already adequately embedded in the model. However, when the training accuracy is lower, the label trick can increase both the training and test accuracy significantly (once the variance from the dataset divisions is taken into account).
>
> Moreover, for large-scale graphs it may be less likely that high accuracy or overfitting is obtained during training due to the large number of training samples.  So in such cases, the label trick is more likely to be effective.
>
> *Table 4*: Although a few results are comparable to those without the label trick, our main goal is to present a novel use case and to show that C&S is parameterizable. However, we have only used the simplest linear model here, which has been able to achieve decent results. This motivates its application to more complex cases, including nonlinear models, where regular C&S cannot perform.
>
> Regardless, the most important contribution of our paper is not promoting a new technique per se, but rather, to provide a theoretical foundation for the label trick, while demonstrating some previously-untried applications of the label trick that can motivate further use cases.
>
> **Question:** (2) Following from the previous point, I would like to get more clarity on the role of label trick in GNN training. The study makes a very shallow remark that: "For example, at the time of this submission, the top 10 results spanning multiple research teams all rely on the label trick, as do the top 3 results from the recent KDDCUP 2021 Large-Scale Challenge MAG240M-LSC." That is okay, however, it does not mean that the label trick necessarily leads to better performance. To be able to argue that, one would need to run an ablation study comparing top models from the Challenge with and without the label trick. Are there other desirable effects that made the label trick so popular (e.g., faster convergence, more robust training etc.)?
>
> **Response:** To the best of our knowledge, there has thus far been no demonstration that the label trick leads to faster convergence or more robust training, and currently, it has only been applied and motivated as a simple means of improving final predictive accuracy.  Note also that, while not all leaderboard submissions or competition entries focus on ablations, there does already exist convincing published work that clearly demonstrates the empirical advantages of the label trick in improving accuracy (which is our focus) over a variety of equivalent base models without the label trick.  For example, please see references Wang et al., "Bag of tricks for node classification with graph neural networks," DLG-KDD Workshop 2021, Shi et al., "R-UniMP: Solution for KDDCUP 2021 MAG240M-LSC," arXiv 2021, Shi et al., "Masked Label Prediction: Unified Message Passing Model for Semi-Supervised Classification," IJCAI 2021, for relevant ablations.

---

### Official Review · Reviewer_RzBD · 2021-11-02

**Correctness:** 4
**Technical Novelty And Significance:** 4
**Empirical Novelty And Significance:** 3
**Recommendation:** 8
**Confidence:** 2

**Main Review:**

The work is solid with rigorous formulation, theoretical justification, and novel use cases of the label trick. The authors derive general formulations for the label trick as regularizations in both regression and classification cases. The derivation is presented in a smooth way and provides several inspiring and insightful remarks and conclusions. The proposed use cases based on the theoretical are interesting and promising according to the empirical evaluations. Overall, I recommend acceptance due to the thorough analysis and potentially broader use cases inspired by the work. However, my assessment might be inaccurate due to my limited expertise in this topic.

**Summary Of The Paper:**

The authors provide a thorough theoretical justification to the recently emerging and heuristically proposed label trick, i.e., adopting a random portion of node labels in the training set as the input node attributes. Specifically, the authors start from the simplest case of label trick and derive the general formulation of the label trick as a regularizer. Motivated by the theoretical analysis, the authors proposed novel use cases of the label trick in broader scenarios.

**Summary Of The Review:**

A solid and inspiring theoretical work. Novel use cases are introduced based on theoretical grounding and are justified by promising empirical results.

---

> ### Author Response · Authors · 2021-11-18
> **Response to Reviewer RzBD**
>
> Thanks for the careful reading, and the positive points the reviewer has highlighted directly correspond with our original motivation for this work. We now also have a revised version which further improves the paper significantly, including a new technical result (Theorem 3) that deals with the $\alpha \rightarrow 1$ limit in general nonlinear scenarios (e.g., GNNs).

---

### Decision · Program_Chairs · 2022-01-20

**Decision:**

Accept (Poster)

**Comment:**

The paper provides the theoretical justification for the "label trick" (using labels in graph-based semisupervised learning tasks). The authors performed a thorough evaluation of their analysis, which constitutes an experimental contribution. The authors provided a rebuttal that the AC finds to have reasonably addressed the reviewers' concerns. We recommend acceptance.